# Multiplex bead assays enable integrated serological surveillance and reveal cross-pathogen vulnerabilities in Zambezia Province, Mozambique

Andrea C. Carcelen [1]✉, Celso Monjane[2], Sophie Bérubé[3], Saki Takahashi [4], Thebora Sultane[2], Imelda Chelene[2], Gretchen Cooley[5], E. Brook Goodhew[5], Catriona Patterson [6], Kevin Tetteh[6], Manuel Mutambe[2], Melissa M. Higdon [1], George Mwinnyaa[1], Gilberto Nhapure[7], Pedro Duce[7], Diana L. Martin[5], Christopher Drakeley [6], William J. Moss[1,4] & Ivalda Macicame[2]

Multiplex serological assays simultaneously measure antibodies to multiple antigens, furnishing insights into exposure and susceptibility to several pathogens and cross-pathogen vulnerabilities. Our serosurvey tests dried blood spots from 1292 individuals for IgG antibodies to 35 antigens from 18 pathogens using a multiplex bead assay for vaccine preventable diseases, malaria, SARS-CoV-2, neglected tropical diseases, and enteric pathogens in Mozambique. We produce pathogen-specific seroprevalence estimates and age-seroprevalence curves and identify spatial differences in seroprevalence. Rural clusters have higher odds of seropositivity to most NTDs neglected tropical diseases, *Plasmodium falciparum* malaria, and enteric pathogens, but lower odds of seropositivity to SARS-CoV-2 and vaccine preventable diseases compared to urban clusters. This co-occurrence identifies clusters with high vulnerability to multiple pathogens. We identify a candidate group of antigens that are correlated with high overall vulnerability. Our results demonstrate a role for multiplex serology in integrated disease surveillance to guide control strategies for individual and co-endemic pathogens.

Serological surveillance—measuring IgG antibodies—allows for the estimation of population level exposure and susceptibility, complementing vaccine coverage data and case-based, wastewater, and genomic surveillance[1]. Serological data provide a measure of past exposure to pathogens and response to vaccination, and allow for assessment of how exposure and susceptibility vary across time, space, age, and other characteristics[2]. These measurements provide accurate estimates of population exposure, which are difficult to obtain with traditional surveillance methods that rely on identifying active infections. Antibody responses known to be relatively short-lived after

[1]Department of International Health, International Vaccine Access Center, Johns Hopkins Bloomberg School of Public Health, Baltimore, MD, USA. [2]Instituto Nacional de Saúde, Maputo, Mozambique. [3]Department of Biostatistics, The University of Florida, Gainesville, FL, USA. [4]Department of Epidemiology, Johns Hopkins Bloomberg School of Public Health, Baltimore, MD, USA. [5]Division of Parasitic Diseases and Malaria, Centers for Disease Control and Prevention, Atlanta, GA, USA. [6]Department of Infection Biology, London School of Hygiene and Tropical Medicine, London, UK. [7]Instituto Nacional de Estatística, Maputo, Mozambique. ✉e-mail: acarcel1@jh.edu

pathogen exposure can be used to quantify recent infections without the need to actively identify cases. The long duration of other antibody responses allows measurement of epidemiologically important quantities such as force of infection or changes in transmission over time, all with the ease of a cross-sectional sample.

While the epidemiological interpretation of seropositivity varies across antigens (e.g., past exposure, recent exposure, or immunity), IgG serological data has multiple public health applications such as assessing disease burden, targeting control efforts, measuring the impact of interventions, characterizing immunity gaps for vaccine-preventable diseases (VPDs), identifying and tracking transmission of emerging and re-emerging infectious diseases, and verifying disease elimination[3]. Serosurvey data are particularly valuable when case-based surveillance or vaccine coverage data are suboptimal[2].

The utility of serosurveillance is extended by serological assays that use multiplex technologies to simultaneously measure antibodies to dozens of target antigens from the same or different pathogens in a single sample. Multi-pathogen serosurveillance is increasingly used to complement disease surveillance systems and achieve economies of scale by maximizing the information from a single specimen, consistent with the goals of collaborative surveillance, which aims to break down siloed disease surveillance systems and replace them with an integrated system across diseases[4].

Additional public health use cases arise from understanding the co-endemicity of pathogens using multiplexed data. First, identifying pathogens that tend to co-occur in individuals, particularly due to shared risk factors (e.g., mosquito-borne pathogens), informs interventions that most effectively target a group of pathogens with a common mode of transmission (e.g., vector control). Second, multiplexed serological studies identifies population-level spatial heterogeneities in transmission across pathogens, highlighting areas that have a higher burden of disease due to diverse groups of pathogens with different modes of transmission. This information could be used to design and target integrated control strategies (e.g., insecticide-treated bed net distributions bundled with vaccination campaigns). Finally, multi-pathogen serosurveillance could be used to characterize overall vulnerability to disease (e.g., areas with limited access to healthcare or clean water and sanitation), thereby enabling horizontal interventions such as infrastructure improvement to be deployed more strategically.

We conducted a multiplexed serosurvey of 35 antigens from 18 pathogens (Table 1) spanning 5 disease categories (VPDs, enteric pathogens, malaria, SARS-CoV-2, and neglected tropical diseases (NTDs) in Zambezia Province, Mozambique by leveraging a population-based registration system, the Countrywide Mortality Surveillance for Action (COMSA)[5]. The primary aim was to quantify and describe exposure patterns to infectious diseases of public health importance and examine multi-pathogen exposures in rural and urban settings. We first computed seroprevalence measures by pathogen. We then extended the analysis beyond individual pathogens to estimate associations between seropositivity to groups of pathogens, including a cluster-based analysis to identify spatial heterogeneity in the seroprevalence of groups of pathogens and in the overall vulnerability to these 18 pathogens. Finally, we identified candidate sentinel antigens that were associated with cluster-level overall vulnerability to these pathogens.

## Results

### Participant characteristics

The survey enrolled 1409 participants, with dried blood spot (DBS) specimens collected from 98% (N = 1383). After excluding specimens that did not pass quality control due to data collection errors such as contamination or laboratory issues such as low bead counts, the final analytic sample comprised 1292 specimens (Fig. 1). The median age of participants was 10 years, with 432 participants aged 6–59 months, 439 aged 5–17 years, and 421 aged 18–49 years. 53% of participants were female, and 86% of participants lived in rural clusters (Table 2).

### Seroprevalence to VPDs

Seroprevalence to VPD antigens were computed using external correlates of immunity. Higher seroprevalence to these pathogens can be interpreted as lower vulnerability to future outbreaks, regardless of whether this immunity was acquired through vaccination or natural infection. Overall measles seroprevalence was 66% (95% CI 64–69%) (Fig. 2A), well below the 90 to 95% needed for herd protection[6]. Rubella seroprevalence was 81% (95% CI 79–83%), increased with age, and primarily represents natural infection since rubella vaccine was introduced in 2018 (Fig. 2B). Seroprevalence to measles was lower among males (63%; 95% CI 59–67%) than females (70%; 95% CI 66–73%) (Fig. 2C). A sensitivity analysis lowering the measles seropositivity threshold from 153 to 120 mIU/mL, a commonly used cutoff[7], eliminated this sex difference (Supplementary Table 3). Participants residing in urban clusters had higher measles and tetanus seroprevalence than those in rural areas (Fig. 2D). Seroprevalence was heterogenous across clusters, ranging from 34 to 98% for tetanus, 57–96% for diphtheria, 64–96% for rubella, and 41–86% for measles (Supplementary Fig. 6). Clusters with low seroprevalence to one VPD antigen tended to have low seroprevalence to multiple VPDs, consistent with poorer performing immunization systems.

### Seroprevalence to Plasmodium species

Zambezia is known to be an area of high *Plasmodium falciparum* transmission[8]. Seroprevalence to *P. falciparum* antigens reflecting past exposure over longer time frames was high: 87% (95% CI 85–89%) for PfAMA1, 67% (95% CI 64–69%) for PfGLURPR2, and 66% (95% CI 64–69%) for PfMSP119. Antibody prevalence to these antigens increased with age (Fig. 3A). The estimated force of infection (FOI) for *P. falciparum* ranged from 0.07 (95% CI 0.06–0.08) for CSP to 0.09 per year (95% CI 0.08–0.10) for PfGLURPR2 implying on average that 7–9% (i.e., 1-exp(-FOI)) of seronegative individuals seroconverted each year (Supplementary Fig. 9). Antigens representing more recent exposure had lower seroprevalence: 1% (95% CI 1–2%) for PfGexp18-2 and 14% (95% CI 12–16%) for PfRh4.2. Seroprevalence was low to *P. vivax* (1%; 95% CI 0–1%; consistent with the lack of documented *P. vivax* transmission in Mozambique[8], *P. ovale* (3%; 95% CI 2–4%), and *P. malariae* (4%; 95% CI 3–5%) species-specific MSP119 protein antibodies. Females had slightly higher seroprevalence to PfMSP119 (70%; 95% CI 66–73%) than males (62%; 95% CI 58–66%). As expected, rural clusters had higher seroprevalence to all *P. falciparum* antigens than urban clusters (Fig. 2D).

### Seroprevalence to SARS-CoV-2

This serosurvey was conducted during the first year of the global COVID-19 pandemic. At the time of this serosurvey, SARS-CoV-2 seroprevalence was only 3% (95% CI 2–4%) to the receptor binding domain (RBD) and 6% (95% CI 5–8%) to the nucleocapsid (N) protein. Seroprevalence to SARS-CoV-2 was higher in urban than rural areas for both antigens: RBD seroprevalence was 7% in urban clusters (95% CI 4–13%) vs. 2% in rural clusters (95% CI 1–3%), and N protein seroprevalence was 12% in urban clusters (95% CI 7–18%) vs. 5% in rural clusters (95% CI 4–7%) (Fig. 2D).

### Seroprevalence to enteric pathogens

While the two enteric pathogens included in this study, *Cryptosporidium parvum* and *Giardia lamblia*, have been detected in various studies in Mozambique more broadly their burden in Zambezia is largely unknown[9,10]. Seroprevalence to *Cryptosporidium parvum* antigens cp23 and cp17 were 69% (95% CI 66–71%) and 90% (95% CI 88–91%). Seroprevalence to cp17 was greater than 85% across all ages but cp23

**Table 1 | Antigen characteristic list**

| Pathogen | Antigen | Abbreviation | Rationale | Description | MFI cutoff | Cutoff determination method | Sensitivity in our study | Specificity in our study | Source of antigen | Tag | Coupling concentration (ug Ag/1.25e7 beads) | Coupling pH | References |
|---|---|---|---|---|---|---|---|---|---|---|---|---|---|
| Brugia malayi | SXP-1 | Bm14 | Exposure | Seroconversion to positive may take years even in an area of high ongoing transmission and there is evidence that antibodies are long-lived but will eventually go down following cure. | 106.5 | Mean + 3Stdev (CDC controls) | | | Recombinant CDC | GST | 120 | pH 7.2 | 39,40 |
| Brugia malayi | Pepsin inhibitor analog AP-1 | Bm33 | Exposure | Seroconversion to positive may take years even in an area of high ongoing transmission and there is evidence that antibodies are long-lived but will eventually go down following cure. | 375.2 | Mean + 3Stdev (CDC controls) | | | Recombinant CDC | GST-His | 20 | pH 6+2M Urea | 41,42 |
| Chlamydia trachomatis | Hypothetical, T3SS substrate | CT694 | Historical exposure | Provides information about infection, exposure, cumulative infection | 108 | ROC | 96% | 97% | Recombinant CDC | GST | 30 | pH 7.2 | 43 |
| Chlamydia trachomatis | pCT03 ORF | pgp3 | Historical exposure | Provides information about infection, exposure, cumulative infection | 212 | ROC | 93% | 100% | Recombinant CDC | GST | 120 | pH 7.2 | 36 |
| Cryptosporidium parvum | GP40/17-kDa antigen | Cp17 | Exposure | Provides information about previous infection | 72.5 | ROC (Maximum Youden's J) | 100% | 95% | Recombinant CDC | GST | 6.8 | pH 5 | 44–46 |
| Cryptosporidium parvum | 27-kDa antigen | Cp23 | Exposure | Provides information about previous infection | 830 | ROC (Maximum Youden's J) | 100% | 100% | Recombinant CDC | GST | 12.5 | pH 5 | 45–47 |
| Diphtheria | Diphtheria toxoid | Dip tox | Immunity | Vaccine acquired immunity | 123.7 | 0.01 IU, 5PL | | | List Labs (Campbell, CA) | None | 60 | pH 5 | 28,48 |
| Expression tag control | Glutathione S-tranferase | GST | Control | Correct for background reactivity due to GST-tag | 27 | n/a negative control | n/a | n/a | CDC | GST | 15 | pH 5 | 49 |
| Expression tag control | Glutathione S-tranferase | GST | Control | Correct for background reactivity due to GST-tag | 93 | n/a negative control | n/a | n/a | LSHTM | GST | 58 | pH 7.2 | |
| Giardia lamblia | Variant-specific surface protein AS8 | VSP3 | Exposure | Provides information about previous infection | 72 | ROC (Maximum Youden's J) | 100% | 85% | CDC | GST | 60 | pH 5 | 50 |
| Giardia lamblia | Variant-specific surface protein 42e | VSP5 | Exposure | Provides information about previous infection | 148 | ROC (Maximum Youden's J) | 91% | 100% | CDC | GST | 60 | pH 5 | 50 |
| Measles virus | whole virus | wMeV | Vaccination or historical exposure | Vaccine acquired immunity or natural infection. | 215 | 153 mIU, 5PL | | | Zeptometrix (Buffalo, NY) | None | 150 | | 29,51 |
| Onchocerca volvulus | Ag16 Diagnostic antigen | OV16 | Historical exposure | Marker of current or past infections. | 422 | ROC (Maximum Youden's J) | 91% | 95% | Tom Nutman (National | GST | 30 | pH 7.2 | 52 |

## Table 1 (continued) | Antigen characteristic list

| Pathogen | Antigen | Abbreviation | Rationale | Description | MFI cutoff | Cutoff determination method | Sensitivity in our study | Specificity in our study | Source of antigen | Tag | Coupling concentration (ug Ag/ 1.25e7 beads) | Coupling pH | References |
|---|---|---|---|---|---|---|---|---|---|---|---|---|---|
| | | | | Antibody responses take at least 15 months to develop, so not immediate marker of infection. Detectable for several years after infections clear | | | | | Institutes of Health, Bethesda, MD) | | | | |
| Plasmodium falciparum | Circumsporozoite protein full length | CSP | Liver stage exposure/ protection/ RTS,S vaccination | Marker of recent exposure to infective mosquito bite; exposed to host immune system during journey to liver through skin delivery; validated | 392.5 | Maximum negative (LSHTM control) | | | Gennova | None | 2.18 | pH 7.2 | 22,53 |
| Plasmodium falciparum | Early transcribed membrane protein 5 antigen 1 (N-terminal) | Etramp5 Ag1 | Recent exposure | Marker of recent exposure based on blood stage infection; transmembrane protein exposed to immune system on iRBC surface and presumably following destruction; validated | 168 | Maximum negative (LSHTM control) | | | Recombinant LSHTM | GST | 65 | pH 7.2 | 22,25,54-56 |
| Plasmodium falciparum | Gametocyte exported protein 18 | Gexp18 | Recent exposure | Marker of recent exposure based on blood stage infection; transmembrane protein exposed to immune system on iRBC surface and presumably following destruction; validated | 632 | Maximum negative (LSHTM control) | | | Recombinant LSHTM | GST | 26.6 | pH 7.2 | 22,25,56,57 |
| Plasmodium falciparum | Glutamate rich protein region 2 | GLURPR2 | Historical exposure | Marker of historical exposure based on blood stage; validated | 160.5 | Maximum negative (LSHTM control) | | | Michael Theisen | His | 0.084 | pH 7.2 | 22,25,56,58 |
| Plasmodium falciparum | Apical membrane antigen 1 | PfAMA1 | Historical exposure | Marker of historical exposure based on blood stage infection; released from apical tip during RBC invasion. Essential to invasion process; validated | 143 | Maximum negative (LSHTM control) | | | Recombinant LSHTM | His | 7.7 | pH 7.2 | 22,25,56,59,60 |
| Plasmodium falciparum | 19 kDa fragment of merozoite surface protein 1 molecule | PfMSP119 | Historical exposure | Marker of historical exposure based on blood stage infection; major protein on surface of mz. Involved in invasion. Heterogeneous interaction with other merozoite | 405.9 | FMM (LSHTM controls) | | | Recombinant LSHTM | GST | 103 | pH 7.2 | 22,25,56,60-65 |

**Table 1 (continued) | Antigen characteristic list**

| Pathogen | Antigen | Abbreviation | Rationale | Description | MFI cutoff | Cutoff determination method | Sensitivity in our study | Specificity in our study | Source of antigen | Tag | Coupling concentration (ug Ag/1.25e7 beads) | Coupling pH | References |
|---|---|---|---|---|---|---|---|---|---|---|---|---|---|
| | | | | proteins. Full function still to be determined. validated | | | | | | | | | |
| Plasmodium falciparum | Reticulocyte Binding Protein-Like Homologue 4 | Rh4.2 | Medium term exposure and protection | Marker of recent exposure based on blood stage infection; part of invasion machinery. validated | 448 | Maximum negative (LSHTM control) | | | Recombinant LSHTM | GST | 85 | pH 7.2 | 22,66,67 |
| Plasmodium malariae | 19 kDa fragment of merozoite surface protein 1 molecule | PmMSP119 | Historical exposure | Marker of historical exposure based on blood stage infection; based on Pf data | 370 | Maximum negative (LSHTM control) | | | Recombinant LSHTM | GST | 1.15 | pH 7.2 | 68,69 |
| Plasmodium ovale | 19 kDa fragment of merozoite surface protein 1 molecule | PoMSP119 | Historical exposure | Marker of historical exposure based on blood stage infection; based on Pf data | 373.5 | Maximum negative (LSHTM control) | | | Recombinant LSHTM | GST | 9.41 | pH 7.2 | 68 |
| Plasmodium vivax | Duffy binding protein RII | PvDBPRII | Exposure | Vivax invasion machinery component. Duffy binding protein/ligand. validated | 534.3 | Maximum negative (LSHTM control) | | | Chetan Chitnis, Pasteur | GST | 2.23 | pH 7.2 | 70-72 |
| Plasmodium vivax | 19 kDa fragment of merozoite surface protein 1 molecule | PvMSP119 | Historical exposure | As for Pfalciparum. validated | 550 | Maximum negative (LSHTM control) | | | LSHTM | GST | 380 | pH 7.2 | 68 |
| Plasmodium vivax | Reticulocyte-binding protein 2b | PvRBP2b | Exposure | Reticulocyte binding protein homologue. Part of invasion machinery. Validated | 577 | Maximum negative (LSHTM control) | | | Chetan Chitnis, Pasteur | GST | 4.13 | pH 7.2 | 70,73,74 |
| Rubella virus | Whole virus | wRuV | Vaccination or historical exposure | Vaccine acquired immunity or natural infection. | 255 | 9.36 IU, 5PL | | | Commercial | None | 30 | pH 7.2 | 29,51 |
| SARS-CoV-2 Wuhan/L strain | Nucleocapsid protein | SARS2 NP | Exposure | Usually natural infection except in places were nucleocapsid is in vaccines. | 614.8 | Maximum negative (LSHTM control) | | | COVID-19 Protein Portal | His | 3.4 | pH 7.2 | 75,76 |
| SARS-CoV-2 Wuhan/L strain | Spike glycoprotein S1 receptor-binding domain | SARS2 RBD | Vaccination or natural exposure | Vaccine acquired immunity or natural infection. | 387.5 | Maximum negative (LSHTM control) | | | Native Antigen Company | His | 15 | pH 7.2 | 75,76 |
| Schistosoma mansoni | Soluble egg antigen | SEA | Historical exposure | Measures antibodies against scistosome eggs. | 410 | ROC (Maximum Youden's J) | 85% | 98% | CDC | None | 120 | pH 7.2 | 77,78 |
| Schistosoma mansoni | | Sm25 | Historical exposure | Measures antibodies against S. mansoni adult worms. | 38.5 | ROC (Maximum specificity) | 75% | 98% | Recombinant CDC | GST | 12 | pH 7.2 | 78-80 |
| Strongyloides stercoralis | | NIE | Exposure | High titers can be indicative of current chronic infection if | 526 | ROC (Maximum Youden's J) | 81% | 100% | Recombinant CDC | GST | 20 | pH 7.2 + 2 M Urea | 81,82 |

**Table 1 (continued) | Antigen characteristic list**

| Pathogen | Antigen | Abbreviation | Rationale | Description | MFI cutoff | Cutoff determination method | Sensitivity in our study | Specificity in our study | Source of antigen | Tag | Coupling concentration (ug Ag/ 1.25e7 beads) | Coupling pH | References |
|---|---|---|---|---|---|---|---|---|---|---|---|---|---|
| | | | | individuals have never been treated. | | | | | | | | | |
| Taenia solium | T. solium excretory secretory 33 kDa | ES33 | Exposure | Antibodies are markers of exposure to tape worm. | 22 | ROC (Maximum specificity) | 88% | 100% | Recombinant CDC | GST | 10 | pH 5 | 83 |
| Taenia solium | Cysticercosis marker | T24H | Exposure | Antibodies are markers of exposure to cysts. | 103.5 | ROC (Maximum specificity) | 86% | 100% | Recombinant CDC | GST | 120 | pH 5 | 84,85 |
| Tetanus | Tetanus toxoid | Tet tox | Immunity | Vaccine acquired immunity. | 30.6 | 0.01 IU, 5PL | | | List Labs (Campbell, CA) | None | 12.5 | pH 5 | 30,86 |
| Treponema pallidum pertenue | Yaws | rp17 | Historical exposure | Current infection or previous exposure to T. pallidum subsp pallidum (syphilis) or T. pallidum subsp pertenue (yaws). Marker of historical infection. | 73.5 | ROC (Maximum Youden's J) | 94% | 95% | ViroGen (Boston, MA) | Beta galactosidase | 15 | pH 7.2 | 27 |
| Treponema pallidum pertenue | Treponemal membrane protein A (Yaws) | TmpA | Recent exposure | Marker of current infection to T. pallidum subsp pallidum (syphilis) or T. pallidum subsp pertenue (yaws). Responses decline after treatment. | 247 | ROC (Maximum Youden's J) | 100% | 96% | ViroGen (Boston, MA) | Beta galactosidase | 30 | pH 7.2 | 27,87 |
| Cercopithecus aethiops | Uninfected Vero cell lysate | UnLys | Control | Identify and eliminate samples with high background for cell line used to produce viral antigens wMeV and RuV. | n/a | n/a negative control | n/a | n/a | CDC | n/a | 150 | 7.2 | 15 |
| Wuchereria bancrofti | Larval antigen | Wb123 | exposure | Antibodies expressed by the L3 larval stage transmitted by mosquitoes | 58.2 | Mean +3Stdev (CDC controls) | | | Tom Nutman (National Institutes of Health, Bethesda, MD) | GST | 30 | pH 7.2 | 88–90 |

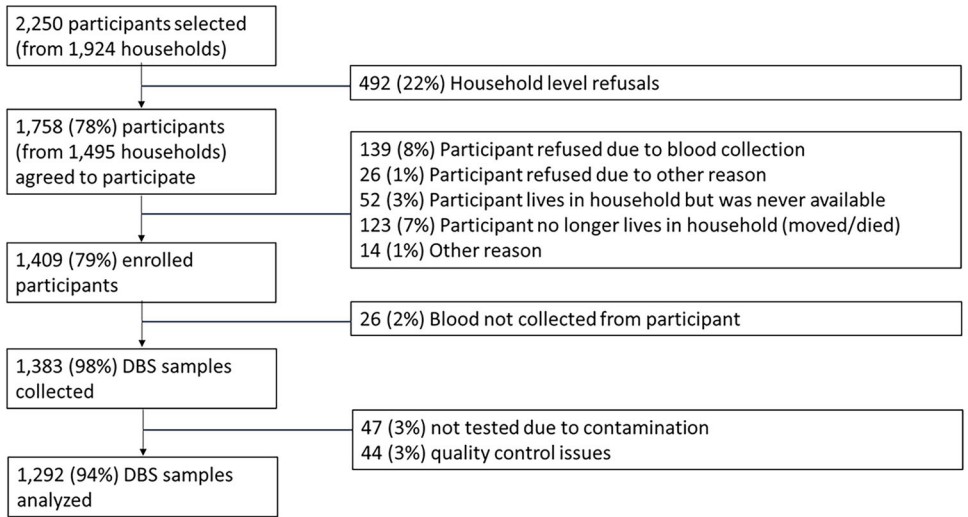

**Fig. 1 | Participant enrollment cascade.** Participants were selected at the individual level irrespective of household. However, during enrollment, some heads of households refused to have anyone in their household participate. Therefore, 22% of participants were lost due to household level refusals. The rest were due to individual level reasons. Quality control issues with the specimens included data collection and laboratory issues such as contamination or low bead counts. Percentages are calculated based on the previous step in the participant cascade.

seroprevalence increased with age (Fig. 3B). Seroprevalence to cp17 was higher in rural areas (91%; 95% CI 89–93%) than urban areas (82%; 95% CI 75–87%) (Fig. 2D). Cluster seroprevalence ranged from 30–91% for cp23 and 60%–100% for cp17, with a mean seroprevalence of 68% and 89% respectively. Giardia seroprevalence was highest for children younger than 5 years, with vsp3 seroprevalence 56% (95% CI 51–61%) and vsp5 41% (95% CI 36–46%). Seroprevalence to vsp5 was one of the only antigens to decrease with age (Fig. 2B). Giardia seroprevalence was higher in rural areas than urban areas for both vsp3 (53% vs. 32%) and vsp5 (24% vs. 14%) (Fig. 2D).

### Seroprevalence to NTDs

This panel included 13 antigens from 8 NTD pathogens with various exposure routes and vectors. Previous estimates of seroprevalence to NTDs in Zambezia were unknown or outdated. Among children 1–9 years of age, the target age range for trachoma control programs and for which seropositivity reflects ocular *C. trachomatis* exposure, pgp3 seroprevalence was 35% (95% CI 31–39%) and CT694 seroprevalence was 38% (95% CI 34–42%). Seroprevalence to CT694 was three times higher in rural clusters (30%; 95% CI 26–35%) compared to urban clusters (8%; 95% CI 4–18%) in this age group. Although overall seroprevalence differed by sex (Fig. 2C), this was not observed in the 1–9 year old age group (females 20% 95% CI 16–26% vs. males 15% 95% CI 11–20%).

Seroprevalence to both treponemal antigens (TmpA and rp17) was 6% (95% CI 4–9%) in children 5–14 years of age and 4% (95% CI 2–6%) in children 1–4 years of age. Seroprevalence exceeded 5% in 13 rural clusters among 5–14-year-olds and in 10 rural clusters among 1–4-year-olds.

Seroprevalence was higher to the pan-*Schistosoma spp* SEA antigen (52%; 95% CI 50–55%) and *Schistosoma mansoni* sm25 antigen (46%; 95% CI 43–49%) than the combination (31%; 95% CI 28–33%). Seroprevalence for each antigen increased with age, particularly in children younger than 10 years of age (Fig. 3C). Seroprevalence to sm25 was higher in rural areas at 68% (95% CI 66–71%) than urban areas at 48% (95% CI 41–56%). The mean cluster seroprevalence for SEA was 59% (range 18–87%) and 63% (range 24–91%) for sm25.

Seroprevalence to the *Onchocerca volvulus* antigen Ov16 was 1.5% (95% CI 0.8–2.9%) among children 1–9 years of age and 1.04% (95% CI 0.25–4.12%) for children 5–9 years, the age groups typically monitored

for determining the need for mass drug administration. Cluster seroprevalence was concentrated in four districts, particularly in one cluster in Morrumbala District (Supplementary Fig. 6).

Seroprevalence to the *Taenia solium* antigen ES33 (a probable marker of tapeworm infection) was 6% and 14% to T24H (associated with the presence of cysts). Seroprevalence to T24H was higher in rural (15%; 95% CI 13–17%) than urban areas (8%; 95% CI 4–13%). Seroprevalence to the *Strongyloides stercoralis* antigen NIE increased with age (Fig. 3C) and had little spatial heterogeneity.

Seroprevalence to pan-filarial antigens varied. Seroprevalence to the Wb123 antigen was highest (25%; 95% CI 22–27%), with lower seroprevalence to Bm33 (10%; 95% CI 9–12%) and Bm14 (8%; 95% CI 7–10%). Like other NTDs, seroprevalence was higher in rural than urban areas across all antigens, most notably for Wb123, with 28% in rural areas (95% CI 26–31%) vs 8% (95% CI 5–13%) in urban areas.

### Individual cross-pathogen vulnerabilities

To measure associations between the seroprevalence of different pathogens and identify candidate use cases for integrated control strategies, we compared adjusted odds ratios of individual seropositivity to pairs of antigens (Fig. 4 and Supplementary Table 5). The 82 significant associations were divided into three types: associations where regressor and outcome antigens are from the same pathogen (31/82), associations where the regressor is from the same pathogen group as the outcome antigen but not the same pathogen (20/82) (e.g., VPDs, *Plasmodium* species, enteric pathogens, or NTDs), and associations where regressor antigen is from a different pathogen group than the (31/82). Associations between antigens from the same pathogen are expected, while associations between antigens from the same pathogen group could result from shared factors influencing the risk of exposure or vaccination, cross-reactivity, or a combination of both. For example, individuals seropositive to rubella virus (regressor antigen) had 4.7 (95% CI 3.3–6.5) times the adjusted odds of being seropositive to measles virus (outcome antigen), which is expected given that measles and rubella antibodies are induced from a single combined vaccine. However, individuals seropositive to rubella (regressor antigen) had 1.8 (95% CI 1.3–2.5) times adjusted odds of being seropositive to tetanus toxoid (outcome antigen), suggesting correlations between the receipt of different vaccines. Finally, associations across pathogen categories are unlikely to be the result of cross-reactivity,

**Table 2 | Participant characteristics in final analytic dataset**

| Individual characteristics (N = 1292) | Unweighted% (n) | Weighted % |
|---|---|---|
| Age | | |
| 6–59 months | 33·4 (432) | 26·1 |
| 5–17 years | 34·0 (439) | 38·1 |
| 18–49 years | 32·6 (421) | 35·7 |
| Sex | | |
| Male | 47·4 (612) | 47·1 |
| Female | 52·6 (680) | 52·9 |
| Education | | |
| Adult's education | | |
| No formal education | 29·5 (124) | 34·5 |
| Primary | 41·8 (176) | 46·8 |
| Secondary | 13·8 (58) | 14·6 |
| Higher education | 0·5 (2) | 0·5 |
| No response | 14·5 (61) | 3·6 |
| Children's maternal education | | |
| No formal education | 49·7 (433) | 50·0 |
| Primary | 35·5 (309) | 35·4 |
| Secondary | 7·0 (61) | 7·0 |
| Higher education | 0·5 (4) | 0·4 |
| No response | 7·3 (64) | 7·2 |
| Occupation | | |
| Adult's occupation | | |
| Unemployed | 20·2 (85) | 20·5 |
| Employed | 7·1 (30) | 7·7 |
| Homemaker | 11·2 (47) | 11·2 |
| Student | 2·9 (12) | 2·9 |
| Other | 45·1 (190) | 45·5 |
| No response | 13·5 (57) | 12·1 |
| Children's maternal occupation | | |
| Unemployed | 22·6 (197) | 21·9 |
| Employed | 6·2 (54) | 6·9 |
| Homemaker | 21·5 (187) | 20·6 |
| Student | 2·0 (17) | 1·9 |
| Other | 38·5 (335) | 38·2 |
| No response | 9·3 (81) | 10·5 |
| Household characteristics[a] (N = 1136) | | |
| Head of household sex | | |
| Male | 69·2 (761) | 68·6 |
| Female | 30·8 (339) | 31·4 |
| Head of household education | | |
| No formal education | 30·4 (335) | 30·5 |
| Primary | 46·5 (511) | 47·1 |
| Secondary | 16·3 (179) | 15·5 |
| Higher education | 1·4 (15) | 1·2 |
| No response | 5·4 (60) | 5·8 |
| Head of household occupation | | |
| Unemployed | 24·9 (274) | 25·1 |
| Employed | 11·7 (129) | 12·2 |
| Homemaker | 6·0 (66) | 6·0 |
| Student | 0·2 (2) | 0·2 |
| Other | 57·1 (628) | 56·5 |
| No response | 0·1 (1) | 0·1 |
| Socioeconomic status | | |
| First quintile (Lowest) | 17·6 (193) | 17·2 |
| Second quintile | 11·0 (120) | 11·1 |

**Table 2 (continued) | Participant characteristics in final analytic dataset**

| Individual characteristics (N = 1292) | Unweighted% (n) | Weighted % |
|---|---|---|
| Third quintile | 39·0 (427) | 39·6 |
| Fourth quintile | 22·7 (249) | 22·6 |
| Fifth quintile (Highest) | 9·7 (107) | 9·6 |
| Participants enrolled | | |
| 1 per household | 87·6 (995) | 88·1 |
| 2 per household | 11·1 (126) | 10·6 |
| 3 per household | 1·3 (15) | 1·3 |
| Cluster characteristics (N = 30) | | |
| Location | | |
| Urban | 16·7 (5) | 18·4 |
| Rural | 83·3 (25) | 81·6 |
| Districts | | |
| Alto Molocue | 3·3 (1) | 4·0 |
| Gurue | 10·0 (3) | 10·9 |
| Ile | 6·7 (2) | 5·6 |
| Inhassunge | 6·7 (2) | 5·8 |
| Lugela | 6·7 (2) | 5·6 |
| Maganja da Costa | 3·3 (1) | 4·0 |
| Milange | 10·0 (3) | 7·7 |
| Mocuba | 10·0 (3) | 11·9 |
| Mocubela | 3·3 (1) | 1·9 |
| Morrumbala | 23·3 (7) | 23·4 |
| Namarroi | 3·3 (1) | 4·0 |
| Nicoadala | 6·7 (2) | 7·5 |
| Quelimane | 6·7 (2) | 7·4 |
| Distance to water (mean km, SD) | 135 (79) | 132 (81) |

[a] 36 households missing information

and therefore provide further evidence of shared risk factors such as exposure to insect vectors. Among these, associations between malaria and NTDs were especially prevalent. For example, individuals seropositive to etramp5 (*P. falciparum*) (regressor antigen) had 2.7 (95% CI 1.8–6.2) times the adjusted odds of being seropositive to CT694 (trachoma) (outcome antigen), and individuals seropositive to GLURP2 (*P. falciparum*) (regressor antigen) had 2.1 (95% CI 1.3–3.5) times the adjusted odds of being seropositive to pgp3 (trachoma) (outcome antigen). Malaria is a vector-borne disease transmitted by mosquitos and trachoma is transmitted via flies; therefore, this association could indicate the need for integrated insect control. Those seropositive to CSP (*P. falciparum*) (regressor antigen) had 1.9 (95% CI 1.3–2.9) times the adjusted odds of being seropositive to Wb123 (lymphatic filariasis) (outcome antigen), also a mosquito-borne illness, and those seropositive to MSP119 (*P. falciparum*) (regressor antigen) had 2.0 (95% CI 1.3–3.1) times the adjusted odds of being seropositive to rp17 (yaws) (outcome antigen).

## Spatial cross-pathogen vulnerabilities

Although individual-level cross-pathogen vulnerabilities furnishes insights into pathogens for which integrated control strategies may be impactful, many of these pathogens have clear spatial heterogeneities in prevalence (e.g., urban vs. rural clusters) that may drive cross-pathogen associations. Seroprevalence was higher in rural than urban areas for both individual and combinations of pathogens, highlighting spatial heterogeneities in disease burden identified through multi-pathogen serosurveys. As expected, participants residing in rural clusters had 1.8–5.3 times the odds of being seropositive to *P. falciparum* and some of the NTD antigens due to shared risk factors and

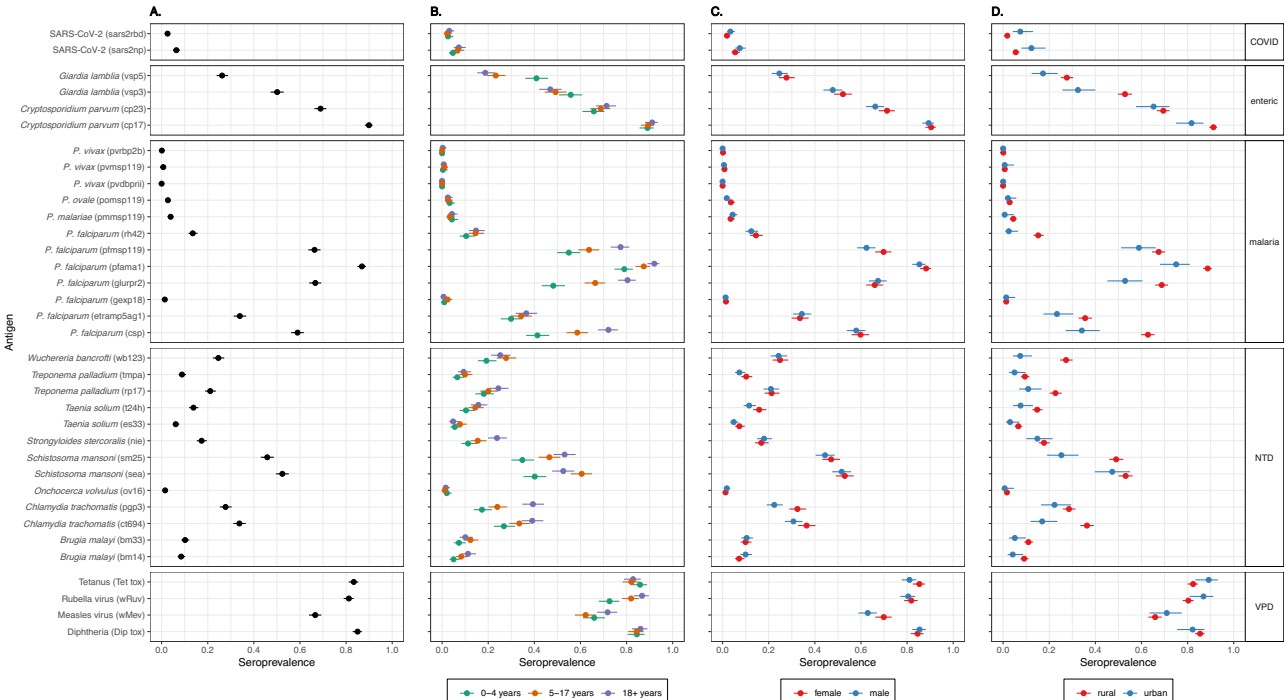

**Fig. 2 | Weighted provincial seroprevalence.** Pathogen with antigen name in parentheses are listed on the *y*-axis; P. is short for *plasmodium*. Weighted provincial seroprevalence is on the *x*-axis. The dot represents mean seroprevalence values and 95% confidence intervals are represented by the lines for each antigen. **A** Black dot represents the provincial estimate (*n* = 1292). **B** Green represents 6month to 4-year-olds (*n* = 432), orange represents 5- to 17-year-olds (439), and purple represents 18- to 49-year-olds (*n* = 421). **C** Red represents seroprevalence for females (*n* = 680), and blue represents males (*n* = 612). **D** Red represents seroprevalence among participants who live in rural clusters (*n* = 1115), and blue represents those in urban clusters (*n* = 177). Source data are in Supplemental Table 4.

vectors. By contrast, seroprevalence estimates for VPDs were lower in rural clusters likely due to lower vaccination coverage (odds ratios for urban vs. rural seroprevalence approximately 2.2 for antigens with significant differences).

To further explore this, we extended our analysis to measure cross-pathogen vulnerabilities, accounting for within-cluster correlations. Using multilevel linear models with cluster-level random intercepts, we computed the odds of seropositivity to 32 antigens (multilevel models for gexp18, etramp5ag1, and rbp2b antigens failed to converge) by cluster relative to a global average, and for each antigen, we ranked clusters from lowest odds of seropositivity to highest odds of seropositivity. Generally, clusters with high odds of seroprevalence for NTDs relative to the global average tended to have higher odds of seroprevalence to *P. falciparum* and lower seroprevalence to VPD antigens, consistent with individual-level findings.

Clusters with the highest overall vulnerability scores were rural clusters (Fig. 5A); however, there was high heterogeneity among these, even at the district level, with Morrumbala District having both the lowest vulnerability clusters (84, 85) and highest vulnerability clusters (79, 83), suggesting a potential role for targeting interventions at fine spatial scales. Beyond being an effective method for identifying specific areas likely to benefit from integrated interventions, we also used overall cluster vulnerability scores to identify whether increased odds of seropositivity to specific antigens were correlated with higher overall disease vulnerability. Of the 32 antigens examined, an increased odds of seropositivity to 11 antigens were significantly correlated with higher vulnerability scores (Fig. 5C). Among these 11 antigens were 2 *P. falciparum* antigens (Pfcsp and PfGLURPR2), a *Giardia* antigen (vsp3), and 8 NTD antigens, many of which were antigens with very low overall seroprevalence such as ov16 from *Onchocerca volvulus* and es33 from *Taenia solium*. These results suggest that low-prevalence diseases,

particularly NTDs, could play an important role in integrated disease surveillance as a sentinel measure of overall vulnerability to a potentially wide range of pathogens.

## Discussion

We used a multiplex bead assay to generate seroprevalence estimates in Zambezia Province, Mozambique for 35 antigens from 18 pathogens spanning 5 disease categories (VPDs, enteric pathogens, SARS-CoV-2, malaria, and NTDs). While some antigens are long-lasting, others may indicate more recent infection (Table 1), therefore changing the interpretation for seropositivity. We identified population-level patterns of seropositivity by age, geography, and other sociodemographic factors. Heterogeneities in exposure between rural and urban settings were identified, particularly for *P. falciparum*, NTDs, and enteric pathogens, as well as vulnerable clusters with a high risk of exposure to multiple pathogens. Seroprevalence estimates filled knowledge gaps where case-based surveillance is less sensitive, such as for low prevalence NTDs, and highlighted population immunity gaps to VPDs, signaling the need for targeted vaccination efforts. Seroprevalence to *P. falciparum* antigens complemented surveillance data by providing estimates of the force of infection consistent with parasite prevalence. Low SARS-CoV-2 seroprevalence was expected because the study period (2020–2021) was early in the pandemic.

This serosurvey provided estimates of exposure to multiple NTDs targeted for elimination, which is particularly valuable where case-based surveillance lacks sensitivity or has suboptimal coverage. Although Mozambique was previously known to be endemic for yaws, the current status is unknown[11]. Our serosurvey found yaws seroprevalence was low overall in Zambezia Province, but there were clusters with seropositive residents that warrant further investigation. Similarly, Mozambique is not considered eligible for preventive chemotherapy for onchocerciasis. However, our seroprevalence estimates

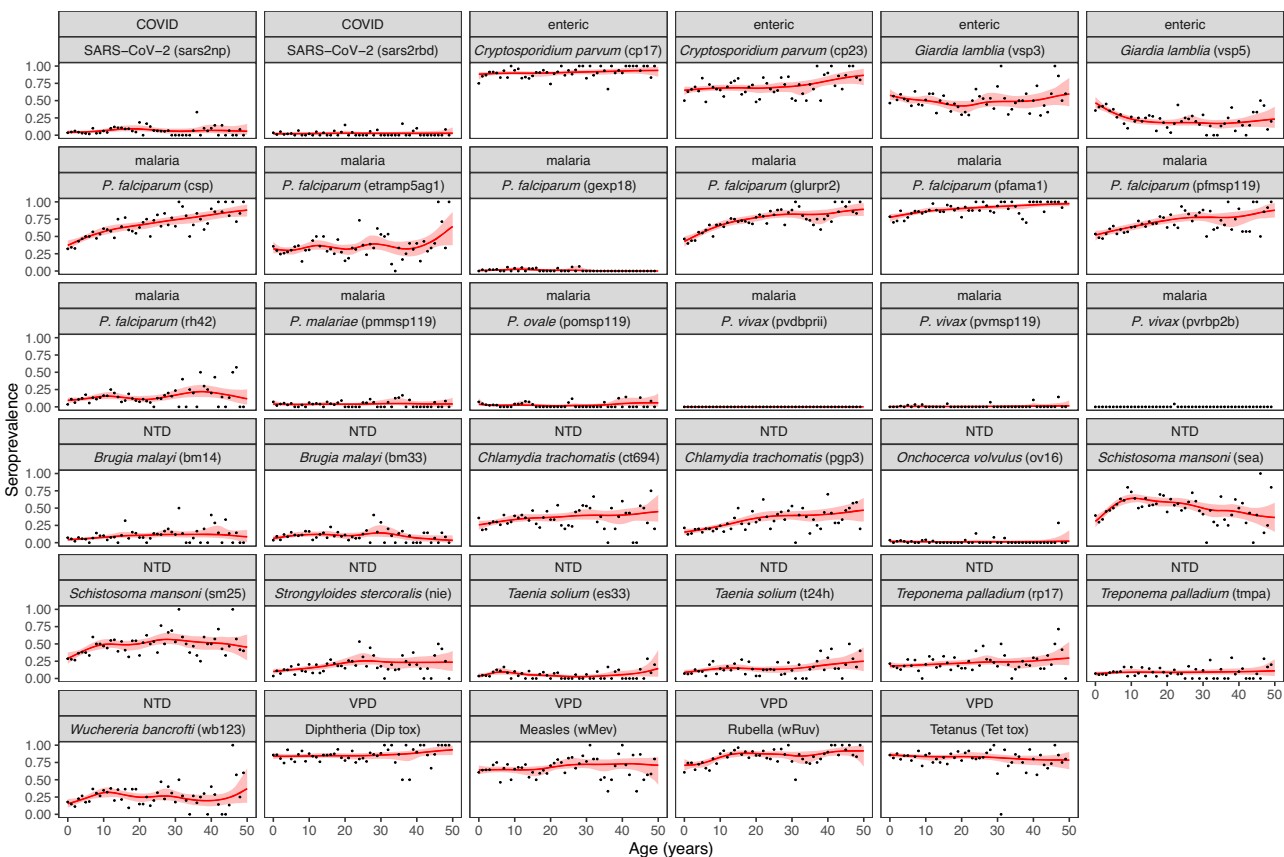

**Fig. 3 | Age seroprevalence curves.** Seroprevalence curves by age for each antigen are represented here. *X*-axis is age in years, and *y*-axis is seroprevalence estimate. Black dots represent observed mean seroprevalence estimates by age. Red lines represent seroprevalence curves by age using semi-parametric cubic regression splines in a generalized additive model (GAM). Red shading represent the 95% confidence intervals for the GAM using bootstrapping. P. vivax (pvrbp2b) failed to converge in calculating confidence intervals for the GAM because there was only 1 seropositive in the dataset. This is based off the seroprevalence responses for all 1292 participants. *P* is short for *plasmodium*, NTD is short for neglected tropical disease, and VPD is vaccine-preventable disease.

exceeded the serological threshold for chemotherapy of 0.1% to OV16 among children 5–9 years of age[12].

Traditional surveillance data sources for malaria (e.g., case surveillance and parasite prevalence surveys) can be biased or limited in their ability to capture underlying *P. falciparum* transmission rates[13]. Although malaria is known to be endemic in Zambezia Province, seroprevalence data for *P. falciparum* antigens that reflect historical exposure in the population allowed estimation of the force of infection. Together with high seroprevalence (>80% to one or more *P. falciparum* antigens) in children younger than 5 years of age, this provides supportive evidence of high levels of malaria transmission, particularly in rural areas. The R21/Matrix-M malaria vaccine was introduced in Zambezia in 2024. We are planning a follow-up serosurvey in 2025 to measure reductions in seroprevalence to short-lived *P. falciparum* antigens to assess the impact of this vaccine.

Seroprevalence data also allowed evaluation of vaccine program performance. Our study found lower seroprevalence to measles than has been found in similar multipathogen studies[14]. According to the Demographic and Health Survey, routine immunization coverage for the first dose of measles-containing vaccine in Zambezia Province was only 35% in 2022[15]. Since this study was conducted, measles outbreaks have been reported in several districts in Zambezia Province[16]. Given the low measles seroprevalence and vaccination coverage, supplemental immunization activities are needed to immunize the one-third of children under five years of age who were measles seronegative. Additional comparisons of seroprevalence with vaccination coverage are planned for future manuscripts.

Leveraging the power of a multiplex assay allowed us to not only investigate pathogen-specific seroprevalence but also assess patterns of disease across pathogens at the individual and cluster levels. Besides the VPDs and SARS-CoV-2, seroprevalence was generally higher in rural areas for both individual and combinations of pathogens, highlighting spatial heterogeneities in disease burden. While other studies have shown similar geographic patterns of seropositivity, they do not combine the seroprevalence results to leverage the results on a multiplex panel[17].

Furthermore, we quantified individual level associations across serostatus and identified associations that could inform integrated control strategies. Our analysis revealed that on an individual level, increased odds of seropositivity to *P. falciparum* antigens was associated with increased odds of seropositivity to a range of NTDs, including those that can be transmitted by mosquitoes or other insect vectors (e.g., lymphatic filariasis and trachoma), suggesting a possible role for integrated insect management.

Given the spatial nature of many risk factors associated with pathogen transmission, and the potential for increased efficacy of interventions when they are spatially targeted, we further extended our analysis using multilevel modeling, accounting for cluster-level associations in seroprevalence. Many of the trends in cross-pathogen vulnerabilities were similar to those identified in the individual-level analysis. For example, many clusters had high odds of seropositivity to both *P. falciparum* and NTD antigens. However, this analysis also enabled us to produce cluster-level vulnerability scores by combining information across all antigens.

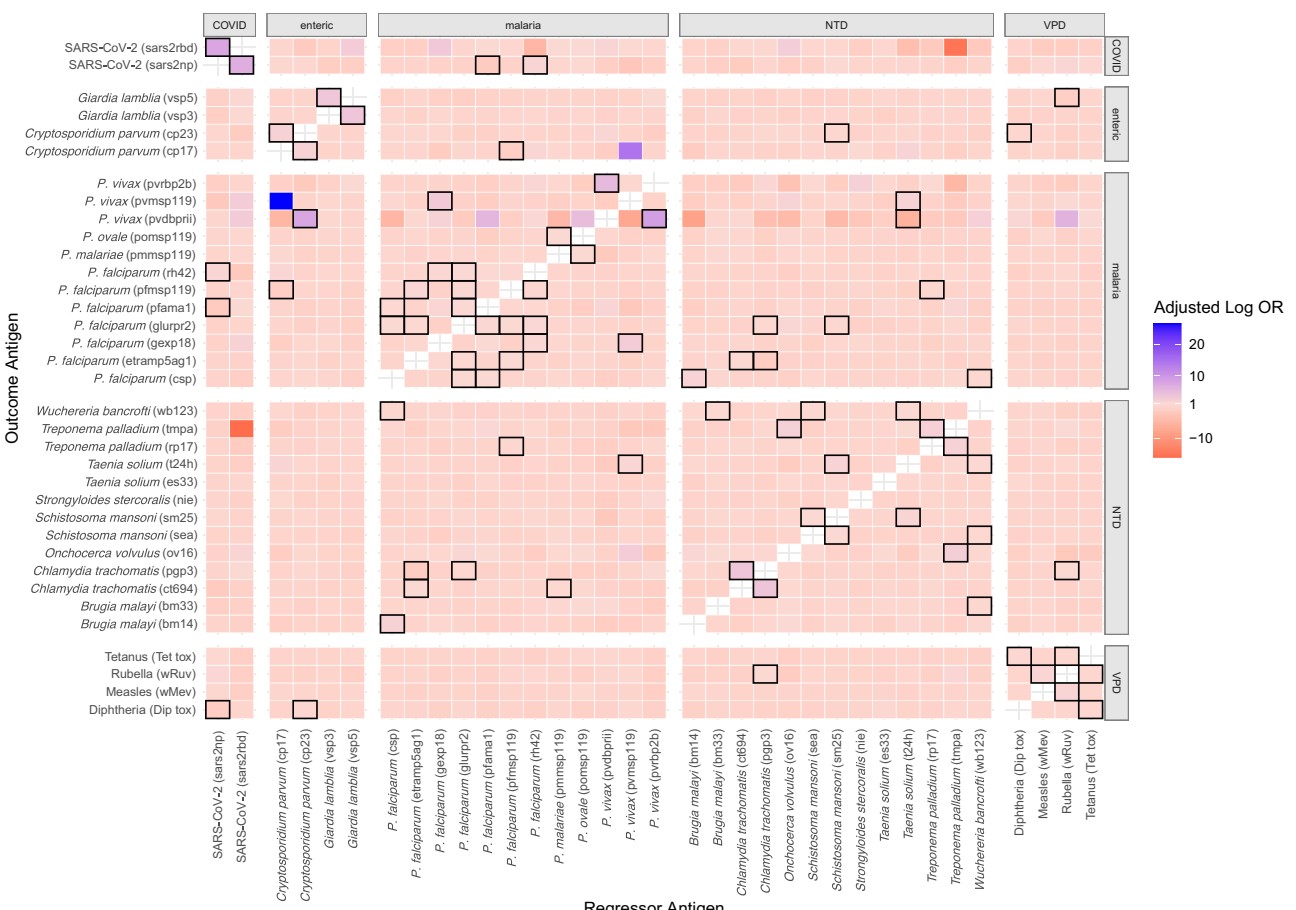

**Fig. 4 | Odds of seropositivity to one antigen compared to another.** Adjusted log odds ratios of seropositivity to outcome antigens on *y*-axis as compared to regressor antigens on *x*-axis among 1292 participants. All odds ratios were calculated using logistic regression and are adjusted for seropositivity to all other antigens, age in years, GST reactivity, and uninfected Vero cell lysate reactivity and urban or rural environment. P. is short for *plasmodium*. Statistically significant ($p < 0.05$ for 2-sided hypothesis test, values adjusted for multiple hypothesis testing) adjusted odds ratios are highlighted in black. Source data are in Supplemental Table 6.

In general, these cluster-level vulnerability scores revealed higher vulnerability in rural compared to urban areas. However, among rural areas, there was high heterogeneity in vulnerability to these pathogens even within the same district, such that Morrumbala District had both the highest and lowest vulnerability clusters. This result supports a role for spatially targeted integrated interventions at the subdistrict level and illustrates the utility of multiplexed serology in identifying these vulnerable areas. These could include providing bednets as part of other health campaigns such as vaccination; water, sanitation, and hygiene interventions alongside mass drug administration for NTDs; or ensuring sufficient health system infrastructure in these areas.

Finally, we identified associations between cluster-level vulnerability and odds of seropositivity to specific antigens. This analysis revealed that exposure to 11 antigens was statistically significantly and positively correlated with high overall vulnerability. Of these, 8 antigens were from NTDs, many with low overall seroprevalence. While low-prevalence diseases are difficult to detect via traditional case-based surveillance, our analysis demonstrates that serology can quantify low-level disease burden and that seropositivity to low-prevalence pathogens may be a useful metric of overall disease burden to multiple pathogens. Many pathogens included in the serosurvey share epidemiological factors associated with increased risk of exposure (e.g., increased exposure to vectors due to housing structure or reduced access to sanitation and healthcare services). Therefore, it is plausible that evidence of low-prevalence pathogens associated with

these risk factors signal exposure to other pathogens with shared risk factors. While more research is required to assess whether these findings generalize to other contexts, multiplexed serological data including low-prevalence sentinel pathogens offer promise as a key component of integrated disease surveillance.

This study has several important limitations. First, for some antigens, there were challenges determining the threshold for seropositivity, as the distributions of antibodies were unimodal, requiring external positive and negative controls (Supplementary Fig. 1). Our cutoffs tended to optimize specificity over sensitivity, which could lead to underestimation of past population exposure. For antigens with known assay sensitivity and specificity, crude seroprevalence estimates and estimates adjusted for test performance were similar, except for *Schistosoma mansoni*, where accounting for low sensitivity of the selected cutoff (75%) resulted in substantially higher seroprevalence estimates (Supplementary Table 2). Second, the assay was optimized for inclusion of multiple targets, so there may be less than optimal laboratory conditions for individual targets. This multiple-antigen approach allows for identification of areas where more targeted screening for specific pathogens could be performed. Third, while all p-values were adjusted to account for the high volume of hypothesis testing, this does not preclude the possibility of spurious correlations, particularly since this is a single observational study confined to one region. Replication studies are required across a range of geographical areas to confirm these associations and to assess their generalizability outside of Zambezia

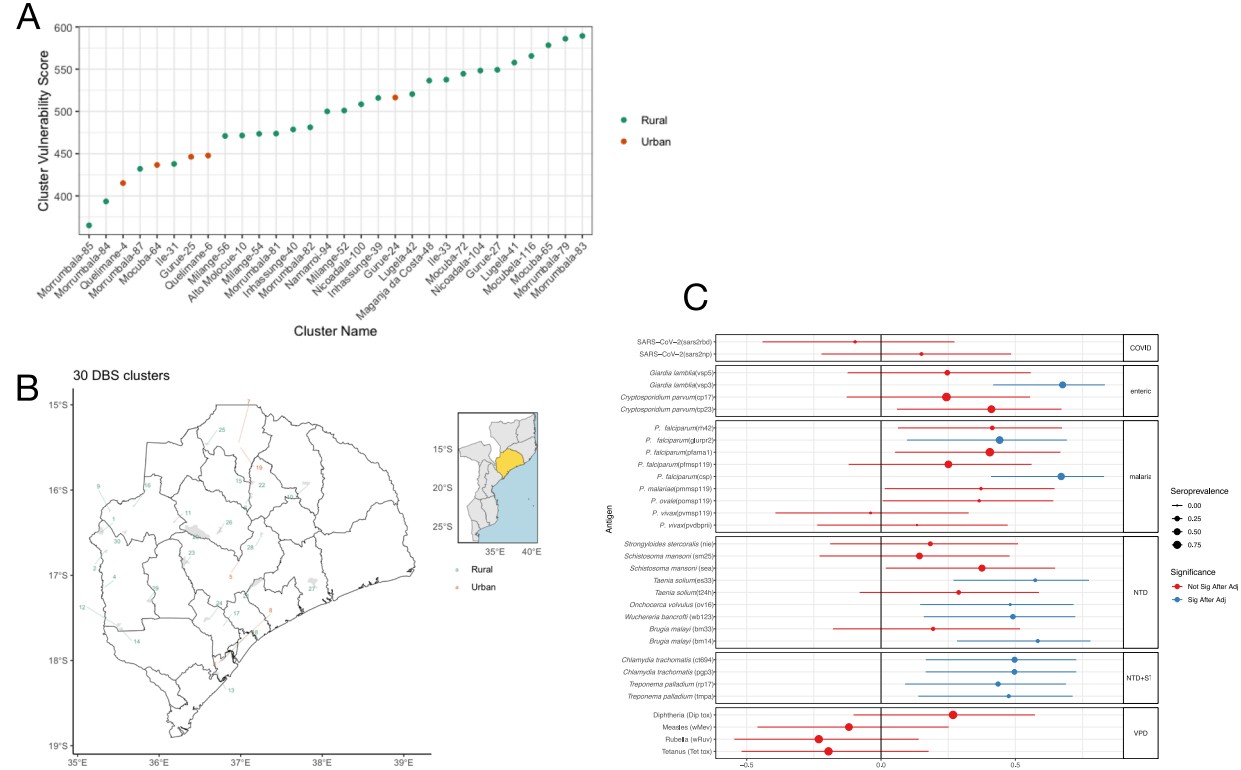

**Fig. 5 | Overall vulnerability score by cluster. A** Vulnerability scores for each cluster (*n* = 30), urban clusters in orange and rural clusters in green. **B** Vulnerability scores for each dried blood spot (DBS) cluster (*n* = 30) ordered from highest (value of 30) to lowest (value of 1) plotted on map of Zambezia province, urban clusters in orange and rural clusters in green Zambezia province within Mozambique shown in yellow inset. This map uses the shape files by United Nations Office for the Coordination of Humanitarian Affairs (OCHA) licensed under CC BY-IGO and available at https://data.humdata.org/dataset/cod-ab-moz. Shape files were not modified.

Source data are in Supplemental Table 4. **C** Spearman's rank correlation and corresponding 95% confidence intervals between rank of odds of seropositivity for each antigen across all clusters (*n* = 30) and rank of overall vulnerability score across all clusters. All *p*-values were adjusted for multiple hypothesis testing using a Benjamini–Hochberg correction. *P*-values that were smaller than 0.05 for a 2-sided *t*-test after adjustment are colored in blue other correlations are colored in red. Size of dots are proportional to the estimated seroprevalence for each antigen. *P* is short for *plasmodium*.

Province. Fourth, for spatially focal pathogens, such as some NTDs, the sample size and sampling frame of our study may have been insufficiently granular to accurately quantify the burden of these pathogens in this region. Relatedly, while our overall pathogen vulnerability estimates are adjusted for age, our sample size precluded estimation of vulnerability stratified by age group, which could further inform the targeting of interventions. Lastly, though we did not see any strong evidence of correlation either between raw median fluorescence intensity (MFIs) across pairs of antigens (Supplementary Fig. 10) or between raw MFI and age among seropositive individuals (Supplementary Fig. 11), suggesting that greater overall probability of exposure to all pathogens was not contributing to higher MFI values and false positive classifications, it is still possible that cross-reactivity could produce false positive results. The three lymphatic filariasis antigens (Wb123 of *Wuchereria bancrofti*, which occurs in Africa; and Bm33 and Bm14 of *Brugia malayi*, which is transmitted in South and Southeast Asia), are known to cross-react. Our non-zero seroprevalence estimates to Bm33 and Bm14 are likely attributable to cross-reactivity. Any further cross-reactivity would result in higher seropositivity, suggesting more past exposure.

Multi-pathogen serosurveillance, along with case-based, wastewater, and genomic surveillance, provides a platform for integrated disease surveillance. Understanding the co-endemicity of pathogens allows for strategies to target interventions to areas with the highest burden. We demonstrate not only how multiple diseases can be monitored for various public health programs through a single serosurvey, but also how seroprevalence estimates can be leveraged

across pathogens to better understand where overall vulnerabilities lie both spatially and across control programs. Further research into how these results generalize to other regions, both within Mozambique and in other areas with potentially differing disease burdens, would expand our understanding how multiplex serology can be leveraged to provide insights into vulnerabilities to a wide range of infectious pathogens.

## Methods

### Study setting
Zambezia Province in the central coastal region of Mozambique is mostly rural with a population of 5.1 million[18]. In 2022, only 66% of the population had access to safe drinking water, and the province had the lowest proportion of fully immunized children 12–23 months of age (13%) and the highest proportion of children who had not received any childhood vaccination (35%), according to the Demographic and Health Survey[15].

### Survey design
From December 2020 to March 2021, we conducted a cross-sectional, population-based serosurvey among individuals 6 months to 49 years of age in Zambezia Province, Mozambique using the COMSA sampling frame to estimate provincial-level seroprevalence to selected VPDs, malaria, NTDs, enteric pathogens, and SARS-CoV-2. COMSA is a community-based surveillance system with provincial representation to provide continuous demographic data on mortality across all age groups[19]. The COMSA platform included 118 sampling clusters in

Zambezia Province, and 30 clusters were selected using weighted stratified (rural/urban) systematic random sampling for the serosurvey, giving higher weight to clusters with larger populations (Supplementary methods). Study participants were selected by simple stratified random sampling within three age groups: 6–59 months, 5–17 years, and 18–49 years. In 18 of 30 clusters, 23 participants were selected per age group, but in 12 clusters, 28 individuals were selected per age group using an adaptive sampling design to account for increased missingness in earlier clusters. In total, 2250 participants were selected from 1924 households. A questionnaire was administered to participants on tablets using ODK software to collect demographic data and risk factors for pathogen exposure. A household-level questionnaire was administered to collect sociodemographic information.

### Ethical approvals
The study protocol was approved by the National Bioethics Committee for Health of Mozambique and the Johns Hopkins Bloomberg School of Public Health Institutional Review Board. Written informed consent for each participant, parental permission for children younger than 18 years of age, and written assent for children 12–17 years of age were obtained. Serological results were not provided to participants, but eligible participants were offered rapid diagnostic tests for HIV and malaria to assess active infection and receive treatment (Supplementary methods).

### Specimen collection and storage
Blood was collected by finger or heel prick on Whatman 903 Protein Saver cards (GE Healthcare, Chicago, IL), dried overnight, and individually stored in plastic bags with desiccants and a humidity indicator card. DBS cards were labeled with a unique participant identification number. DBS cards were transported to the central laboratory at the Instituto Nacional de Saúde and stored at −20 °C until shipment to the Centers for Disease Control and Prevention in Atlanta, GA, USA for testing.

### Multiplex bead coupling and assay
The chemical coupling of antigens to MagPlex microspheres (beads) (Luminex Corp., Austin, TX) was optimized, and antigens were coupled to specific bead regions as previously described with coupling concentrations indicated in Table 1[20–22]. Glutathione-S-transferase (GST) coated beads were included to assess potential non-specific binding to GST-tagged proteins, and uninfected Vero cell lysate coated beads were included as a negative control for viral antigens produced in cell culture. A single batch of beads was coupled for all experiments to avoid lot-to-lot variability. (Supplementary methods). Samples were diluted to a final estimated serum concentration of 1:400 by eluting a 3 mm DBS punch in 500 μL of Buffer B (1× PBS, 0.5% casein, 0.5% polyvinyl alcohol, 0.8% polyvinylpyrrolidone, 0.3% Tween-20, 0.02% sodium azide, and 3 μg/mL Escherichia coli extract) at 4 °C overnight. All incubation steps were performed at room temperature in 50 μl reaction volumes protected from light while shaking at 600 rpm. Each incubation was followed by three washes of 200 μl PBS pH 7.2 containing 0.05% Tween-20 using a 405-TSRS automated plate washer with a magnetic adapter (BioTek, Winooski, VT, USA). Diluted sera were incubated with ~1250 microspheres/antigen/well for 90 min, followed by a 45-min incubation with secondary antibodies diluted to 50 ng/well for anti-human IgG and 40 ng/well for anti-IgG4 (Southern Biotech, Birmingham, AL) in Buffer A (1× PBS, 0.5% BSA, 0.05% Tween-20, and 0.02% NaN3). Plates were then incubated for 30 min with 250 ng/well streptavidin-R phycoerythrin (Invitrogen, Waltham, MA, USA) diluted in Buffer A followed by an incubation in Buffer A alone for 30 min to remove loosely bound antibodies. Finally, microspheres were resuspended in 100 μl PBS pH 7.2 and stored overnight at 4 °C prior to reading on a MAGPIX bioanalyser (Luminex Corporation, Austin, TX, USA)[23].

### Seropositivity thresholds
Data were output as MFI. To control for background reactivity, each assay included blank wells containing Buffer B only. The threshold for seropositivity for each antigen was calculated using one of the following approaches: (1) threshold calculated as the mean plus three standard deviations of the MFI distribution of previously characterized or presumed negative controls from the US (CDC) or UK (LSHTM) (Supplementary Figs. 2 and 4)[24,25]; (2) highest MFI value for negative controls (Supplementary Fig. 3)[26]; (3) receiver operating characteristic (ROC) curve of MFI values for previously characterized negative and positive controls with a threshold that maximized Youden's J-index to balance sensitivity and specificity, or that maximized specificity with a lower bound of 75% for sensitivity (Supplementary Fig. 5)[27]; or (4) concentrations were calculated for VPDs based on linear regression fit to log-transformed values from a dilution series of international standards with seroprotective thresholds based on previously validated international unit (IU) values[28–30]. A primary approach was selected for each antigen, either following the approach used in prior analyses involving these antigens[31] or based on the controls available for that antigen (Supplementary Table 1 and Supplementary Fig. 1).

### Data analysis and modeling
Weighted seroprevalence estimates for each antigen were generated as the proportion of samples equal to or above the threshold for seropositivity, using the individual-level sampling weights. Seroprevalence estimates were stratified by age group, sex, and cluster type (rural or urban), and 95% confidence intervals (CIs) were obtained using binomial errors. Antigen-specific seroprevalence curves by age were constructed using semi-parametric cubic regression splines in a generalized additive model (GAM)[32,33]. For antigen thresholds calculated using the ROC method, adjusted seroprevalence was calculated based on sensitivity and specificity (Supplementary Fig. 5)[34]. For *P. falciparum* antigens known to be markers of historical exposure (PfAMA1, PfMSP119, CSP, and GLURPR2), we used simple serocatalytic models to estimate the force of infection from each age-seroprevalence curve, excluding 0–2 year-olds to allow for decay of maternal antibodies[35]. For *Chlamydia trachomatis* and *Treponema pallidum*, seroprevalence in older age groups may represent sexually transmitted infections (urogenital chlamydia and syphilis, respectively)[27,36]. Thus, serostatus as a proxy of trachoma and yaws exposure is only interpreted for children younger than 15 years. Similarly, some pathogens use seroprevalence in different age groupings to guide programmatic action, such as onchocerciasis. These age groupings are clarified in the data analysis.

To investigate individual-level associations between seropositivity to different antigens, 35 logistic regression models were fit. In each model the outcome was an individual's serostatus (positive or negative) to one of the antigens, and the regressors were individual serostatus to the remaining antigens, along with age, cluster type (rural/urban), MFI values in response to control antigens (GST and uninfected Vero cell lysate) were included to control for non-specific binding[21]. Statistically significant associations were identified using p-values adjusted for multiple hypothesis testing using the method described by Benjamini and Hochberg[37].

To compare the odds of seropositivity in each COMSA cluster relative to the whole survey population, Bayesian logistic random effects models were fit. For each model, estimates of posterior distributions for all cluster random intercepts were extracted and ranked using a squared error loss function[38]. Ranks were then averaged across all posterior draws. This information was combined into an overall cluster vulnerability score by taking the sum of all antigens ranks within a cluster, with higher ranks for malaria, NTDs, enteric pathogens, and SARS-CoV-2 contributing positively to the score and higher ranks for VPDs contributing negatively to the score, assuming thresholds for seropositivity are correlates of protective immunity.

Thus, if a cluster had a higher overall vulnerability score, individuals in this cluster were more likely to be at higher risk of pathogen exposure. Finally, the correlation between overall cluster vulnerability scores and the cluster-specific ranks of each antigen were computed using Spearman's rank correlation coefficient. Statistically significant correlations were identified using *p*-values adjusted for multiple hypothesis testing using the method described by Benjamini and Hochberg[37]. Additional details in Supplemental methods.

Analyses were performed using STATA (version 14.2), SAS (version 9.4), and R (version 4.2.2).

## Reporting summary

Further information on research design is available in the Nature Portfolio Reporting Summary linked to this article.

## Data availability

Raw individual-level data that underlie the results reported in this article (text, tables, figures, and supplementary material) and the study protocol are protected and are not available publicly due to data privacy laws. These data are available after de-identification upon request to the National Institute of Health of Mozambique via the corresponding author beginning one year after publication and ending three years after publication. Researchers must submit a proposal to be approved by authorities in Mozambique using an institutional data request form. The processed data denoting overall seroprevalence, seroprevalence by age, sex, geography, and cluster are provided in Supplementary Information. Source data for Figs. 2 and 5A are in Supplementary Table 4, and source data for Fig. 4 are available in Supplementary Table 6.

## Code availability

R code for this analysis is available at https://github.com/sberube3/COMSA_moz_analysis.git.

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

## Acknowledgements

We would like to acknowledge the COMSA team members, including Agbessi Amouzou, Malick Kante, Akum Aveika, and Fred Van Dyk at Johns Hopkins University and Azarias Mulungo, Nordino Machava, and Victor Maive at Instituto Nacional de Saude. We thank the COMSA study team for their tireless efforts to collect data during the COVID-19 pandemic and participants for their contributions to public health. This work was funded by the Bill and Melinda Gates Foundation grants (#029405 and #060109). The findings and conclusions in this report are those of the authors and do not necessarily represent the official position of the Centers for Disease Control and Prevention.

## Author contributions

A.C.C., C.M., W.M.J., and I.M. contributed to the design and implementation of the research. C.M., T.S., I.C., M.M., and G.N. were involved in planning and supervising data collection. W.M.J., I.M., and P.D. ensured ethical approvals and provided oversight. G.C., E.B.J., D.L.M., C.P., K.T., and C.D. provided the multiplex bead assay panel, conducted laboratory testing, aided in interpreting the results, and worked on the manuscript. M.H. and G.M. supported supply procurement, training, and contributed to data analysis. S.B. and S.T. conducted cross-pathogen analyses and designed the figures. C.C., S.B., S.T., and W.M.J. conducted the analyses and the writing of the manuscript.

## Competing interests

The authors declare no competing interests.

## Ethical approval

The research included local researchers throughout the research process. The research was determined in collaboration with local partners and has been presented to the local authorities. Roles and responsibilities were agreed among collaborators ahead of the research. This research was not severely restricted or prohibited in the setting of the researchers. Any risk of stigmatization, incrimination, discrimination, or personal risk to participants was minimized through local ethical considerations, including compliance with anonymity, privacy, and referrals to health facilities as dictated by local Ministry of Health policies. The study protocol was approved by the National Bioethics Committee for Health of Mozambique and the Johns Hopkins Bloomberg School of Public Health Institutional Review Board.
