## [Peer Review file · Nature Communications]

Multiplex bead assays enable integrated serological surveillance and reveal cross-pathogen vulnerabilities in Zambezia Province Mozambique

Corresponding Author: Dr Andrea Carcelen

Version 0:

Reviewer comments:

Reviewer #1

(Remarks to the Author)

The articles present the results of serological surveillance for a range of infections in Mozambique using a multiplex bead assay. The study is generally well-conducted, providing additional insights of infection dynamics in a rural area and low income country with high levels of disease endemicity.

Comments:

1) The authors do not adequately put their findings in context of the results from previous serological surveillance studies or surveys of disease prevalence in the region. How do the findings compare with studies in other areas of sub-Saharan Africa or other areas? A very cursory search reveals several other studies that are not discussed or compared: Njenga, Sammy M. et al. "Integrated cross-sectional multiplex serosurveillance of IgG antibody responses to parasitic diseases and vaccines in coastal Kenya." *The American journal of tropical medicine and hygiene* 102, no. 1 (2019): 164.

Gwyn, Sarah, et. al "Performance of SARS-CoV-2 antigens in a multiplex bead assay for integrated serological surveillance of neglected tropical and other diseases." *The American Journal of Tropical Medicine and Hygiene* 107, no. 2 (2022): 260.

Fornace, Kimberly M., et al. "Characterising spatial patterns of neglected tropical disease transmission using integrated sero-surveillance in Northern Ghana." *PLoS neglected tropical diseases* 16, no. 3 (2022): e0010227.

2) The specific VPDs and NTDs included should be mentioned earlier in the text or included in a table for easier reference

3) Some additional detail is needed regarding assay development. Line 143- which proteins were GST tagged? Where were positive controls for ROC curves obtained and how were they verified (citation only refers to one paper referencing yaws). Were MFI values background subtracted?

4) MFI responses to IgG can increase with age due to multiple infections with the same or similar pathogens and due to cross-reactivity with other infections. This can increase false positives as age increases. Please provide the correlations between MFI and age (not the association between seroprevalence and age) and discuss this issue

5) The authors note that some of their targets do not show evidence of a bimodal distribution- this is likely because some of the infections studied (Crypto) are acute, IgG responses are not long - lasting and individuals can have multiple infections over time. This makes seroprevalence estimates not very meaningful, a higher MFI may indicate a more recent infection. While the authors mention this early in the paper, they still discuss seroprevalence for these acute infections Some discussion of which infections studied have lifetime or long-lasting IgG antibody responses and which wane soon following infection is warranted.

6) Line 213- clarify the reasons for excluding samples and what QC criteria were

7) Line 227- were concentrations estimated? Should be described in the methods or supplement

8) Line 331 - "Low SARS CoV-2" this sentence is not clear, the low prevalence could be because it was early during the pandemic, I don't think the low seroprevalence suggests the need to monitor ongoing virus transmission. Please delete this or clarify

9) Line 415- "Since this study...." citation needed for measles outbreaks statement.

10) For seropositivity determinations, these seem highly subjective: "If sensitivity is lower than desired.."; "If sensitivity, specificity, or seroprevalence do not seem reasonable...", please mention what was deemed reasonable and desired and if any sensitivity analyses were done.

11) In the supplement it seems SAS was used but this is not mentioned in the text

12) The regression models described in 181 may be overfitted and may not be identifiable (i.e., there could be some cells where there is no data) - serostatus to each of the antigens was included in the model resulting in a huge model. Did all models converge and produce valid estimates? Why were MFI values for targets that do not have a GST tag included? Was age included in the model? At least some additional results and details from these regression models (coefficients, p-values, etc.) should be provided in the supplement

13) The p-value correction for false discovery rate is Benjamini and Hochberg, not Hoshberger or Hoshberg, please correct in text and supplement

(Remarks on code availability)

Reviewer #2

(Remarks to the Author)

Carcelen et al sought to use a multiplexed assay to measure antibodies against 35 antigens, derived from 18 pathogens to quantify and describe exposure patterns to infectious diseases of public health importance and examine multi-pathogen exposures in rural and urban settings. The serosurvey spanned 5 disease categories including vaccine preventable diseases (VPDs), enteric pathogens, malaria, SARS-CoV-2 and neglected tropical diseases (NTDs), and identified a candidate group of 11 pathogen antigens whose seropositivity correlated with high overall vulnerability to disease. Authors also identified geographic clusters with high vulnerability to multiple pathogens. The paper is very well written and the conclusions drawn are valid.

The following are however concerns that authors will need to address to enhance the quality and impact of the paper.

1. Sample size - There seems to be some number disagreements between sample sizes stated in the methods and results sections. The methods section mentions 2250 selected participants (page 4) but the results section indicates that 1409 participants were enrolled (page 6). Authors should clarify and reconcile these numbers.

2. Antigen coupling to beads – Authors have described coupling antigens to microspheres, but this section lacks details. What quantity of each of the 35 antigens was coupled to the microspheres? The quantities may not be exactly what are described in the referenced papers.

3. Seropositivity thresholds – Were all the indicated criteria applied to all antigens for defining the thresholds? What informed using four different criteria to define a single threshold for seropositivity for each antigen? If different criteria were used for different antigens, what was the basis for selecting specific criteria for some antigens and other criteria for other antigens?

4. Seroprevalence to VPDs - Was there an attempt to compare antibody responses between vaccinated and un-vaccinated children for VPDs? Also, was there age-stratification in the levels of responses, especially when most vaccines are given to children? Were antibody levels higher for children compared to adults due to vaccination, or rather higher in adults due to possible repeated natural exposure? There also needs to be some discussion of these.

5. Age categories for comparison – different age categories have been used for comparison of seroprevalence data for some of the pathogen-specific antigens (mostly on page 8). While there may be a logical reason for basing comparisons on these age cut-offs, this information is not provided in the manuscript. Authors should provide justification for how the different age categories were used for the said seroprevalence comparisons.

Minor edits

Page 4, line 127 - "was" should rather be "were"

Page 8, line 281 – seroprevalence "was" 38%

(Remarks on code availability)

Reviewer #3

(Remarks to the Author)

Dr Carcelen et al., present findings from an extensive serological dataset, investigating the population seroprevalence of 18 pathogens in Zambezia, Mozambique. The manuscript is well written, with interesting results and well-supported conclusions. It is a very nice use case for the broad utility of multi-pathogen serological surveillance.

My main comment relates to the individual-level cross-pathogen vulnerabilities approach. The methodology of these individual-level models is not fully clear to me. The results mentions 82 significant associations found, though it is not clear from how many models these associations came from. My expectation is that 35 models would be fit (1 for each antigen as the outcome variable) but this should be clarified in the methods.

And if multiple associations can presumably come from the same model, are there duplicates included in this 82 value? For instance, the association between measles and rubella could come from the model where measles is the outcome variable and where rubella is the outcome variable. I understand the intention behind the analysis and the odds ratio results make sense but clarifications on this approach are needed to understand these numbers quoted in the results.

An additional supplementary figure that would be useful is a heatmap of correlation coefficients for antibody responses (MFI levels) between every possible pair of antigens. This would give a more direct measure of cross-reactivity than looking at associations in seropositivity across antigens. This could also potentially strengthen the discussion where the possibility of cross-reactivity biasing results is mentioned.

Minor comments:

- Line 223 regarding measles seroprevalence states "... represents both natural infection and vaccination". When I first read this in the results, I did not know how you could conclude this until I reached the discussion and saw that outbreaks have been reported and vaccine coverage is reportedly much lower. I would suggest rephrasing to either not mention the source of seroprevalence or else provide the citations here in the results to back up the sentence.
- In the discussion regarding correlations in low prevalence NTD vulnerability, it would be great to have a comment on the power of the study design & assay specificity to accurately estimate seroprevalence when the true burdens are very low.

(Remarks on code availability)

Version 1:

Reviewer comments:

Reviewer #1

(Remarks to the Author)

The authors have addressed my previous concerns adequately

(Remarks on code availability)

No issues noted

Reviewer #3

(Remarks to the Author)

The authors have sufficiently addressed the comments, with an improved version of the manuscript.

(Remarks on code availability)

Reviewer #5

(Remarks to the Author)

The responses to the reviewer's comments and the revised manuscript demonstrate a thorough and satisfactory effort to address the concerns raised. Below is an assessment of each comment and the corresponding revisions:

Comment 1: Sample size discrepancy

Concern: Discrepancy between the sample sizes mentioned in the methods (2,250 participants) and results (1,409 enrolled).

Author's Response: The authors clarified the discrepancy by adding a participant flow chart (Figure 1) that details the selection, enrollment, blood collection, and testing stages, explaining losses at each step.

Assessment: The response is clear and resolves the confusion by providing a visual and textual explanation of the sample size reduction. The flow chart in the revised manuscript (Page 18) effectively illustrates the attrition process.

Comment 2: Antigen coupling details

Concern: Lack of details on the quantity of antigens coupled to microspheres.

Author's Response: The authors moved coupling concentrations and pH details from the supplement to Table 1 in the main text, citing specific references for each antigen.

Assessment: The revision is satisfactory. Table 1 (Pages 26–31) now provides comprehensive details, enhancing transparency and reproducibility.

Comment 3: Seropositivity thresholds

Concern: Clarification needed on the criteria used for defining seropositivity thresholds.

Author's Response: The authors explained that the methodology depended on control availability and documented the process in Supplementary Table 1. They also referenced Table 1 for cutoff values.

Assessment: The response is valid. The methods section (Page 5, lines 166–180) and Table 1 provide sufficient detail, though the rationale for selecting specific criteria per antigen could be slightly expanded for clarity.

Comment 4: Seroprevalence to VPDs

Concern: Request for comparison of antibody responses between vaccinated and unvaccinated children and age-stratified analysis.

Author's Response: The authors noted that such comparisons are reserved for a separate manuscript and restricted this analysis to binary seropositivity. They emphasized using seroprevalence to identify vaccination gaps.

Assessment: The response is reasonable given the scope of the current manuscript. However, a brief mention of planned future work in the discussion could further justify this limitation.

Comment 5: Age categories for comparison

Concern: Justification needed for varying age categories in seroprevalence comparisons.

Author's Response: The authors clarified that age groupings were pathogen-specific, guided by programmatic action needs (e.g., NTDs).

Assessment: The explanation is adequate, though a sentence in the methods or results explicitly stating this rationale would strengthen clarity.

Minor Edits

Suggestions: Corrections for grammatical errors ("was" to "were") and seroprevalence percentage.

Author's Response: The errors were fixed.

Assessment: The corrections are accurately implemented in the revised manuscript.

RECOMMENDATIONS FOR FURTHER IMPROVEMENT:

1. Briefly mention the planned VPD comparison study in the discussion to preempt similar queries.
2. Expand the rationale for age categories in the methods section to aid reader understanding.
3. Strengthen discussion of cross-reactivity: Clarify how cross-reactivity may affect NTD seroprevalence estimates.
4. Highlight public health implications: Emphasize how cluster-level vulnerability scores could guide integrated interventions (e.g., bundling bed nets with vaccination campaigns).
5. Minor edits: Ensure consistency in reporting seroprevalence confidence intervals (e.g., "95% CI 64–69%" vs. "95%CI 66–73%").

(Remarks on code availability)

I have checked on the code accessibility/availability; however, I am not able to assess to what extent the results of the paper are reproducible and the code is a usable resource for the community, given that I do not have expertise in coding.

REVIEWER COMMENTS

Reviewer #1 (Remarks to the Author):

The articles present the results of serological surveillance for a range of infections in Mozambique using a multiplex bead assay. The study is generally well-conducted, providing additional insights of infection dynamics in a rural area and low income country with high levels of disease endemicity.

Comments:

1) The authors do not adequately put their findings in context of the results from previous serological surveillance studies or surveys of disease prevalence in the region. How do the findings compare with studies in other areas of sub-Saharan Africa or other areas? A very cursory search reveals several other studies that are not discussed or compared: Njenga, Sammy M. et al. "Integrated cross-sectional multiplex serosurveillance of IgG antibody responses to parasitic diseases and vaccines in coastal Kenya." *The American journal of tropical medicine and hygiene* 102, no. 1 (2019): 164.

Gwyn, Sarah, et. al "Performance of SARS-CoV-2 antigens in a multiplex bead assay for integrated serological surveillance of neglected tropical and other diseases." *The American Journal of Tropical Medicine and Hygiene* 107, no. 2 (2022): 260.

Fornace, Kimberly M., et al. "Characterising spatial patterns of neglected tropical disease transmission using integrated sero-surveillance in Northern Ghana." *PLoS neglected tropical diseases* 16, no. 3 (2022): e0010227.

This is well noted. It was challenging to contextualize all of the pathogens in the discussion, but we agree others that have multiple pathogens should be referenced. We have added the Njenga and Fornace references to the discussion. However, the Gwyn manuscript is primarily about the validation of a particular assay, so we did not include it.

2) The specific VPDs and NTDs included should be mentioned earlier in the text or included in a table for easier reference

This is well noted. The list was included in a supplemental table that has been moved forward as main table 1.

3) Some additional detail is needed regarding assay development. Line 143- which proteins were GST tagged? Where were positive controls for ROC curves obtained and how were they verified (citation only refers to one paper referencing yaws). Were MFI values background subtracted?

The antigens that were GST tagged are indicated in the new table 1. The controls for the ROC curves are noted by the references in Table 1, which lists different references by antigen. We

have moved many of the laboratory details up from the supplement to the main text. Yes, the MFI values were background subtracted using blank wells containing Buffer B only. This was in the supplement, but has been added to the methods section.

4) MFI responses to IgG can increase with age due to multiple infections with the same or similar pathogens and due to cross-reactivity with other infections. This can increase false positives as age increases. Please provide the correlations between MFI and age (not the association between seroprevalence and age) and discuss this issue

We explored correlations between age and continuous MFI across each antigen for all data, seropositive individuals, and seronegative individuals. In general, all correlations were between -0.09 and 0.22 with only seronegative individuals for Rubella showing a higher correlation of 0.31. This is likely due to the fact that rubella vaccination was just introduced in Mozambique in 2018, so most seroprevalence reflects natural infection. Furthermore, considering only seropositive individuals for each antigen (given the stated concern with false positives), only three antigens showed statistically significant correlations with age after adjusting for multiple hypothesis testing: *glurpr2* from *P. falciparum*, *sea* from *Schistosoma mansoni*, and Tetanus. These estimated correlations were very low (between -0.1 and 0.1). From this analysis we cannot conclude that those that were previously classified as seropositive who are older are more likely to have higher MFI and therefore may contribute to false positive classification. We have added a line in paragraph 9 of the discussion with accompanying supplementary figure 11.

5) The authors note that some of their targets do not show evidence of a bimodal distribution- this is likely because some of the infections studied (Crypto) are acute, IgG responses are not long - lasting and individuals can have multiple infections over time. This makes seroprevalence estimates not very meaningful, a higher MFI may indicate a more recent infection. While the authors mention this early in the paper, they still discuss seroprevalence for these acute infections. Some discussion of which infections studied have lifetime or long-lasting IgG antibody responses and which wane soon following infection is warranted.

What each antigen represents (recent infection, exposure, historical exposure, immunity, etc) is noted in the "rationale" column of table 1 with further description provided in the description column. This was originally a supplemental table but has been moved to main manuscript. We have also mentioned this in the discussion.

6) Line 213- clarify the reasons for excluding samples and what QC criteria were. Quality control issues with the specimens included data collection and laboratory issues such as incorrect contamination or low bead counts. We have included the participant enrollment cascade as Figure 1 to clarify the reasons why specimens were excluded at different points.

7) Line 227- were concentrations estimated? Should be described in the methods or supplement

Concentrations were calculated for vaccine preventable diseases only using international standards. This has been clarified in the methodology.

8) Line 331 - "Low SARS CoV-2" this sentence is not clear, the low prevalence could be because it was early during the pandemic, I don't think the low seroprevalence suggests the need to monitor ongoing virus transmission. Please delete this or clarify

Yes, this is primarily due to the time period for this serosurvey being early in the pandemic. This has been clarified.

9) Line 415- "Since this study..." citation needed for measles outbreaks statement.

We have cited the World Health Organization reported measles cases in Mozambique.

10) For seropositivity determinations, these seem highly subjective: "If sensitivity is lower than desired.."; "If sensitivity, specificity, or seroprevalence do not seem reasonable...", please mention what was deemed reasonable and desired and if any sensitivity analyses were done. The cutoff methodology for seropositivity was contingent on the availability of controls. The thought process is documented in Supplementary Table 1, where it is noted that sensitivity needed to be at least 75%. We have updated the table to be clearer.

The actual cutoff value and methodology used for each antigen is captured in table 1. Sensitivity analyses were done but are not presented here. We acknowledge in the limitations section the challenges with determining thresholds for seropositivity and that specificity was prioritized over sensitivity. Because this is not a diagnostic tool, it was not thought necessary to capture every seropositive. Our seroprevalence estimates are thus conservative estimates of past population exposures.

Controls	Methodology	Pick cutoff that
Availability of international standards	Translation of values to international units	corresponds to known correlates of protection
Availability of both positive and negative controls	Receiver Operating Curve	Maximize Youden's J
If sensitivity is low	Receiver Operating Curve with floor value for sensitivity	Maximum specificity possible with sensitivity of at least 75%
If sensitivity, specificity, or seroprevalence do not seem reasonable based on other data sources (e.g. case-based surveillance or previous seroprevalence estimates)	Receiver Operating Curve	Select cutoff to match previous estimates
Availability of only negative controls	Sample mean plus 3 standard deviations	Sample mean plus 3 standard deviations on the natural scale
If controls are not normally distributed or have small number of controls	Highest negative control	
If negative controls have some high values that seem to fit a bimodal distribution	Finite mixture model (2-component model with Gaussian distributions)	Sample mean plus 3 standard deviations on the logarithmic scale
No controls available	Finite mixture model	Sample mean plus 3 standard deviations on the logarithmic scale

11) In the supplement it seems SAS was used but this is not mentioned in the text

Thank you, this has been included in the methods section.

12) The regression models described in 181 may be overfitted and may not be identifiable (i.e., there could be some cells where there is no data) - serostatus to each of the antigens was included in the model resulting in a huge model. Did all models converge and produce valid estimates? Why were MFI values for targets that do not have a GST tag included? Was age included in the model? At least some additional results and details from these regression models (coefficients, p-values, etc.) should be provided in the supplement

The models the author is referring to all achieved convergence (the loss function associated with the iteratively reweighted least squares method of the “glm” function in R was minimized). However, some extremely large odds ratios (e.g. for *P. vivax* (pvmmsp119)) are not significant in these analyses. This is because the confidence intervals for these odds ratios are extremely large and this is reflective of the extremely low seroprevalence for these antigens. We do not take this to be a sign of poor model fit or invalid estimates, but rather of the model accurately reflecting the large uncertainty in this estimate due to the nature of the data. We think these values are still worth reporting while clearly showing that they are not statistically significant as is the case in Figure 4 and while reporting all associated p-values, coefficients and 95% confidence intervals in Supplementary Table 6.

Age was included in all models (see paragraph 2 of the Data analysis and modeling subsection of the methods). GST was included in all models because it has previously been described in other publications (such as Rogier et al) as an indicator of non-specific binding in a sample, this could impact all antigens not just GST tagged antigens. We have mentioned this in paragraph 2 of the Data analysis and modeling subsection of the methods.

Rogier, E., van den Hoogen, L., Herman, C. *et al.* High-throughput malaria serosurveillance using a one-step multiplex bead assay. *Malar J* **18**, 402 (2019). <https://doi.org/10.1186/s12936-019-3027-0>

13) The p-value correction for false discovery rate is Benjamini and Hochberg, not Hoshberger or Hoshberg, please correct in text and supplement
Thank you. This has been fixed.

Reviewer #2 (Remarks to the Author):

Carcelen et al sought to use a multiplexed assay to measure antibodies against 35 antigens, derived from 18 pathogens to quantify and describe exposure patterns to infectious diseases of public health importance and examine multi-pathogen exposures in rural and urban settings. The serosurvey spanned 5 disease categories including vaccine preventable diseases (VPDs), enteric pathogens, malaria, SARS-CoV-2 and neglected tropical diseases (NTDs), and identified a candidate group of 11 pathogen antigens whose seropositivity correlated with high overall

vulnerability to disease. Authors also identified geographic clusters with high vulnerability to multiple pathogens. The paper with very well written and the conclusions drawn are valid. The following are however concerns that authors will need to address to enhance the quality and impact of the paper.

1. Sample size - There seems to be some number disagreements between samples sizes stated in the methods and results sections. The methods section mentions 2250 selected participants (page 4) but the results section indicates that 1409 participants were enrolled (page 6). Authors should clarify and reconcile these numbers.

Thank you for noting this, yes we have added the participant flow chart as Figure 1 to clarify the sample sizes at each time point (from selection to enrollment to blood collection to testing) and how many participants were lost at each. The discrepancies are based on how many participants are selected from the registers of the mortality surveillance system, how many were able to be found when data collection teams went to the field, how many consented to enrollment, how many specimens were viable upon reaching the laboratory, and how many were successfully analyzed and linked back to survey data.

2. Antigen coupling to beads – Authors have described coupling antigens to microspheres, but this section lacks details. What quantity of each of the 35 antigens was coupled to the microspheres? The quantities may not be exactly what are described in the referenced papers. Coupling concentrations and coupling pH for each antigen are included in Table 1 along with specific references for each antigen. This was originally in the supplement, but we moved it up to be Table 1. We have also moved many of the laboratory details up from the supplement to the main text.

3. Seropositivity thresholds – Were all the indicated criteria applied to all antigens for defining the thresholds? What informed using four different criteria to define a single threshold for seropositivity for each antigen? If different criteria were used for different antigens, what was the basis for selecting specific criteria for some antigens and other criteria for other antigens? Only one methodology was used for each antigen. The methodology was contingent on the availability of controls. This has been clarified in the methods section text.

The thought process is documented in Supplementary Table 1 and noted below. This was done in consultation with subject matter experts and informed by previous studies. The actual cutoff value and methodology used for each antigen is captured in the main table 1.

Controls	Methodology	Pick cutoff that
Availability of international standards	Translation of values to international units	corresponds to known correlates of protection
Availability of both positive and negative controls	Receiver Operating Curve	Maximize Youden's J
If sensitivity is low	Receiver Operating Curve with floor value for sensitivity	Maximum specificity possible with sensitivity of at least 75%
If sensitivity, specificity, or seroprevalence do not seem reasonable based on other data sources (e.g. case-based surveillance or previous seroprevalence estimates)	Receiver Operating Curve	Select cutoff to match previous estimates

Availability of only negative controls	Sample mean plus 3 standard deviations	Sample mean plus 3 standard deviations on the natural scale
If controls are not normally distributed or have small number of controls	Highest negative control	
If negative controls have some high values that seem to fit a bimodal distribution	Finite mixture model (2-component model with Gaussian distributions)	Sample mean plus 3 standard deviations on the logarithmic scale
No controls available	Finite mixture model	Sample mean plus 3 standard deviations on the logarithmic scale

4. Seroprevalence to VPDs - Was there an attempt to compare antibody responses between vaccinated and un-vaccinated children for VPDs? Also, was there age-stratification in the levels of responses, especially when most vaccines are given to children? Were antibody levels higher for children compared to adults due to vaccination, or rather higher in adults due to possible repeated natural exposure? There also needs to be some discussion of these.

We have a separate manuscript in progress with comparisons in antibody responses between vaccinated and un-vaccinated children for VPDs. In this manuscript, we restricted to analyses based on binary seropositivity results. Therefore we did not compare quantitative antibody levels by age or vaccination status in this manuscript but note that for the subsequent planned manuscript. Seroprevalence data provides a direct measure of immunity, allowing us to look at vulnerability to infection based on known correlates of protection. In this manuscript we use the seropositivity to VPDs as measures of vulnerability to identify areas that may require further vaccination.

5. Age categories for comparison – different age categories have been used for comparison of seroprevalence data for some of the pathogen-specific antigens (mostly on page 8). While there may be a logical reason for basing comparisons on these age cut-offs, this information is not provided in the manuscript. Authors should provide justification for how the different age categories were used for the said seroprevalence comparisons.

This is correct, some pathogens use seroprevalence in different age groupings to guide programmatic action, particularly the NTDs. We have tried to make more explicit the reasons with the corresponding pathogens.

Minor edits

Page 4, line 127 - “was” should rather be “were”
Thank you this is fixed.

Page 8, line 281 – seroprevalence “was” 38%
Thank you this is fixed.

Reviewer #3 (Remarks to the Author):

Dr Carcelen et al., present findings from an extensive serological dataset, investigating the population seroprevalence of 18 pathogens in Zambezia, Mozambique. The manuscript is well written, with interesting results and well-supported conclusions. It is a very nice use case for the broad utility of multi-pathogen serological surveillance.

My main comment relates to the individual-level cross-pathogen vulnerabilities approach. The methodology of these individual-level models is not fully clear to me. The results mentions 82 significant associations found, though it is not clear from how many models these associations came from. My expectation is that 35 models would be fit (1 for each antigen as the outcome variable) but this should be clarified in the methods.

We have clarified this point in the second paragraph of the Data analysis and modeling subsection of the methods. The reviewer is correct 35 models were fit. Furthermore, we added Supplementary Table 6 that shows all association values, 95% confidence intervals, and p values for these associations.

And if multiple associations can presumably come from the same model, are there duplicates included in this 82 value? For instance, the association between measles and rubella could come from the model where measles is the outcome variable and where rubella is the outcome variable. I understand the intention behind the analysis and the odds ratio results make sense but clarifications on this approach are needed to understand these numbers quoted in the results.

The specific outcome and regressor variables as well as the adjusted components of the model have been clarified in the Individual cross-pathogen vulnerabilities subsection of the results section. However the “duplicate” values are not actually duplicate values because these are adjusted models (as clarified and stated in the second paragraph of the Data analysis and modeling subsection of the methods). The association between measles and rubella, adjusted for all other responses and other covariates is not the same as the association between rubella and measles adjusted for all other responses and other covariates. Importantly depending on the underlying relationships between covariates and the outcome antigens these may not even both be significant associations, though in our models this is often the case.

An additional supplementary figure that would be useful is a heatmap of correlation coefficients for antibody responses (MFI levels) between every possible pair of antigens. This would give a more direct measure of cross-reactivity than looking at associations in seropositivity across antigens. This could also potentially strengthen the discussion where the possibility of cross-reactivity biasing results is mentioned.

We have included a heatmap of the pairwise correlations between all continuous MFIs in the supplement, importantly, these correlations are not adjusted for any factors and therefore the inference we can draw from them is limited. We mention this as an additional sensitivity analysis in the 9th paragraph of the discussion section and show the results in a Supplementary Figure 10. However, the only pairs of antigens that rose above a correlation of 0.5 were antigens from the same pathogens (e.g. *pgp3* and *ct694* both from *C. trachomatis*, *cp23* and *cp17* both from *Cryptosporidium*). There were some correlations between 0.3 and 0.5 that were identified between malaria antigens of different species (e.g. *gexp18* from *P. falciparum* and *pvmosp119* from *P. vivax*) for which cross-reactivity has previously been described in the literature, and finally some antigens across groups, such as malaria and NTDs, in particular (*pvmosp119* from *P. vivax* and *wb123* from *W. bancrofti*). For this last group, the known cross-reactivity of the *vivax* and *falciparum* antigens, combined with the common exposure route of filariasis and malaria through mosquitoes makes it difficult to disentangle possible cross-reactivity from shared exposure risk factors. Combined, these results do not necessarily paint a clear picture of cross reactivity beyond what is already known in the literature and mentioned in the discussion.

Minor comments:

- Line 223 regarding measles seroprevalence states "... represents both natural infection and vaccination". When I first read this in the results, I did not know how you could conclude this until I reached the discussion and saw that outbreaks have been reported and vaccine coverage is reportedly much lower. I would suggest rephrasing to either not mention the source of seroprevalence or else provide the citations here in the results to back up the sentence.

IgG serology for measles cannot distinguish between natural infection and vaccination. This was an attempt to contextualize what serology represented, but has been deleted here to avoid confusion.

- In the discussion regarding correlations in low prevalence NTD vulnerability, it would be great to have a comment on the power of the study design & assay specificity to accurately estimate seroprevalence when the true burdens are very low.

The study was powered to detect seropositivity point estimates by the 3 defined age groups (6-59 months, 5-17 years, 18-49 years) based on expected seroprevalence for measles and malaria. It is noted in the limitations that for spatially focal pathogens, such as some NTDs, the sample size and sampling frame of our study may have been insufficiently granular to accurately quantify the burden of these pathogens in this region. For these NTDs with low burdens (e.g. *Onchocerciasis*), specificity was prioritized over sensitivity, and yet there was still signal of seroprevalence for these. In further analyses, we are using these data for sampling simulations to assess optimal sampling strategies for NTDs with potentially low seroprevalence.

REVIEWERS' COMMENTS

Reviewer #1 (Remarks to the Author):

The authors have addressed my previous concerns adequately

Reviewer #1 (Remarks on code availability):

No issues noted

Reviewer #3 (Remarks to the Author):

The authors have sufficiently addressed the comments, with an improved version of the manuscript.

Reviewer #5 (Remarks to the Author):

The responses to the reviewer's comments and the revised manuscript demonstrate a thorough and satisfactory effort to address the concerns raised. Below is an assessment of each comment and the corresponding revisions:

Comment 1: Sample size discrepancy

Concern: Discrepancy between the sample sizes mentioned in the methods (2,250 participants) and results (1,409 enrolled).

Author's Response: The authors clarified the discrepancy by adding a participant flow chart (Figure 1) that details the selection, enrollment, blood collection, and testing stages, explaining losses at each step.

Assessment: The response is clear and resolves the confusion by providing a visual and textual explanation of the sample size reduction. The flow chart in the revised manuscript (Page 18) effectively illustrates the attrition process.

Comment 2: Antigen coupling details

Concern: Lack of details on the quantity of antigens coupled to microspheres.

Author's Response: The authors moved coupling concentrations and pH details from the supplement to Table 1 in the main text, citing specific references for each antigen.

Assessment: The revision is satisfactory. Table 1 (Pages 26–31) now provides comprehensive details, enhancing transparency and reproducibility.

Comment 3: Seropositivity thresholds

Concern: Clarification needed on the criteria used for defining seropositivity thresholds.

Author's Response: The authors explained that the methodology depended on control availability and documented the process in Supplementary Table 1. They also referenced Table 1 for cutoff values.

Assessment: The response is valid. The methods section (Page 5, lines 166–180) and Table 1 provide sufficient detail, though the rationale for selecting specific criteria per antigen could be slightly expanded for clarity.

Comment 4: Seroprevalence to VPDs

Concern: Request for comparison of antibody responses between vaccinated and unvaccinated children and age-stratified analysis.

Author's Response: The authors noted that such comparisons are reserved for a separate manuscript and restricted this analysis to binary seropositivity. They emphasized using seroprevalence to identify vaccination gaps.

Assessment: The response is reasonable given the scope of the current manuscript. However, a brief mention of planned future work in the discussion could further justify this limitation.

Comment 5: Age categories for comparison

Concern: Justification needed for varying age categories in seroprevalence comparisons.

Author's Response: The authors clarified that age groupings were pathogen-specific, guided by programmatic action needs (e.g., NTDs).

Assessment: The explanation is adequate, though a sentence in the methods or results explicitly stating this rationale would strengthen clarity.

Minor Edits

Suggestions: Corrections for grammatical errors ("was" to "were") and seroprevalence percentage.

Author's Response: The errors were fixed.

Assessment: The corrections are accurately implemented in the revised manuscript.

RECOMMENDATIONS FOR FURTHER IMPROVEMENT:

1. Briefly mention the planned VPD comparison study in the discussion to preempt similar queries.

Thank you; this has been added to the discussion.

2. Expand the rationale for age categories in the methods section to aid reader understanding.

This has been added to the methods section.

3. Strengthen discussion of cross-reactivity: Clarify how cross-reactivity may affect NTD seroprevalence estimates.

Added a sentence in the limitations paragraph of the discussion that mentions cross-reactivity to highlight that any further cross-reactivity that has not been accounted for would result in higher seroprevalence estimates, suggesting more exposure to pathogens than may be true.

4. Highlight public health implications: Emphasize how cluster-level vulnerability scores could guide integrated interventions (e.g., bundling bed nets with vaccination campaigns).

We have added additional examples in the discussion section.

5. Minor edits: Ensure consistency in reporting seroprevalence confidence intervals (e.g., "95% CI 64–69%" vs. "95%CI 66–73%").

Thank you; this has been standardized to "95% CI".

Reviewer #5 (Remarks on code availability):

I have check on the code accessibility/availability; however, i am not able to assess to what extent the results of the paper are reproducible and the code is a usable resource for the community, given that I do not have expertise in coding.